# Chemical imaging delineates Aβ plaque polymorphism across the Alzheimer's disease spectrum

Srinivas Koutarapu[1,16], Junyue Ge [1,16], Maciej Dulewicz [1,16], Meera Srikrishna[1,2], Alicja Szadziewska [1], Jack Wood [1,3], Kaj Blennow [1,4,5,6], Henrik Zetterberg [1,4,7,8,9,10], Wojciech Michno [1,11], Natalie S. Ryan[8,12], Tammaryn Lashley [7,13], Jeffrey N. Savas [14], Michael Schöll [1,2,7,12] & Jörg Hanrieder [1,4,7,12,15] ✉

Amyloid-beta (Aβ) plaque formation in Alzheimer's disease (AD) pathology is morphologically diverse. Understanding the association of polymorphic Aβ pathology with AD pathogenesis and progression is critical in light of emerging Aβ-targeting therapies. In this work, functional amyloid microscopy enhanced by deep learning was integrated with mass spectrometry imaging to delineate polymorphic plaques and to identify their associated Aβ make-up. In both sporadic AD ($n = 12$) and familial AD ($n = 6$), dense-core plaques showed higher levels of Aβ1-40 and N-terminal pyroglutamated Aβx-42 compared to diffuse plaques and plaques in non-demented, amyloid positive individuals ($n = 5$). Notably, a distinct dense-core plaque subtype, coarse-grained plaque, was observed in AD but not in non-demented, amyloid positive patients. Coarse-grained plaques were more abundant in early onset AD, showed increased neuritic dystrophy and higher levels of Aβ1-40 and Aβ3pE-40, an Aβ-pattern similar to cerebral amyloid angiopathy. The correlative chemical imaging paradigm presented here allowed to link structural and biochemical characteristics of Aβ plaque polymorphism across various AD etiologies.

Alzheimer's Disease (AD) is an age associated disorder affecting 12% over the age of 65 and poses an ever increasing societal and socioeconomic challenge[1]. The major pathological hallmarks of AD include the progressive, abnormal accumulation and deposition of beta-amyloid (Aβ) peptides as extracellular plaques and hyperphosphorylated tau protein as neurofibrillary tangles[2]. The most widely accepted hypothesis suggests that Aβ aggregation and plaque formation are critical early events in AD pathogenesis that initiate a damaging cascade[3] of downstream events such as synaptic changes, tau pathology and eventual neurodegeneration[4]. The relevance of Aβ pathology in AD, has seen a recent resurgence following positive results from the Phase III study and FDA approval of the Aβ-targeting antibody lecanemab[5–9] and donanemab[10,11], which both provide support of the amyloid cascade hypothesis[3,12]. However, while clinical trial data for those drugs suggest

that all patients respond with reduction in Aβ not all responded well cognitively or with respect to imaging- and fluid-based biomarkers for pathophysiological pathways downstream of Aβ[5,9–11]. The reason for this differential response could be related to heterogenous Aβ pathology typically observed across patients that encompasses different frequencies for a variety of structural heterogenous Aβ aggregates, including parenchymal plaques, categorized into dense-core plaques (CP) and diffuse plaques (DP)[13–15], as well as cerebral amyloid angiopathy (CAA)[16] (Table 1). In dense-core plaques (CP), also referred to as senile plaques[13,17], a compact, fibrillized amyloid core is surrounded by an outer sphere of diffuse, immature Aβ aggregates[15,18] along with tau paired-helical filament (PHF-1) positive dystrophic neurites that denote those deposits as neuritic plaques[15,19]. Diffuse plaques are characterized by a loose arrangement made of immature, mostly prefibrillar amyloid

**Table 1 | Summary of features associated with various plaque morphotypes**

| Type of Plaque | Description and Features | Prevalence |
|---|---|---|
| Diffuse Plaques (DP) | Lacks a dense core, are usually large (50 µm and more), shows moderate immunoreactivity, amyloid fibre bundles in the neuropil without degenerating neurites and accompanying microglia. Predominantly staining for Aβ1-42 but also Aβx-42 incl. AβxpE-42. No IHC staining for Aβx–40. | Considered as the earlies phenotype of precipitating plaque pathology. Present in young DS, early AD and in cognitively unimpaired amyloid positive brains |
| Cored Plaques (CP) | Characterized by a dense core of amyloid surrounded by a halo/corona of premature fibrils, commonly 30–50 µm. Typically stains intensely with amyloid-binding dyes such as thioflavin and congo red. Stains strongly for Aβ1–40, Aβ1-42 and N-truncated/pyroglutamated Aβ. Levels of N-terminal truncation higher than in diffuse plaques. | Typical for AD and DS brains but also observed in cognitively unimpaired amyloid positive brains |
| Neuritic Plaques (NP) | Presence of dystrophic neurites- swollen neurites originated from axons. Often surrounded by reactive glial cells. Distinguishable by PHF-1 (IHC). | Hallmark of AD pathology associated with local immune activation, neuronal network dysfunction, and cognitive decline. |
| Coarse-Grained Plaques (CGP) | Characterized by dense amyloid deposition, mostly 50–100 µm. Show higher levels of Aβx-40 and less Aβx-42 compared to cored plaques. Linked to a higher degree of neuritic dystrophy, associated with intense neuroinflammation. | Prominent in early onset AD and ApoE4/E4 but also observed in late onset, sporadic AD |
| Cotton Wool Plaques (CWP) | Large, amorphous, and fluffy plaques without a compact amyloid core, found in some familial AD cases. Do not stain with thioflavin-S. | Observed in presenilin-1 mutant cases of familial AD |

aggregates[20–22]. The prevalence of polymorphic Aβ plaques is associated with clinical dementia, where the neocortical load of neuritic plaque correlates with clinical symptoms[23,24]. As diffuse plaques are observed in very early Down's syndrome[25] and are the predominant plaque morphotype in cognitively unimpaired, amyloid-positive[26–28], they are widely considered an initial, pre-mature stage of Aβ deposition that mature into dense-core plaques upon AD progression[29]. The formation of dense-core plaques and eventually neuritic plaques types, presumably evolving from diffuse plaques, is consequently regarded critical in AD pathogenesis[28,29] and understanding the biochemical composition of distinct Aβ plaque morphotypes is therefore essential, particularly in light of amyloid targeting therapies. Previous studies provided valuable information on the N- and C-terminal truncation patterns of Aβ specific to distinct plaque types in situ[25,29–31] despite those studies relying on immuno-based methods that have limitations with respect to chemical specificity[32], sensitivity, quantification and the degree of comprehensive molecular information. Most importantly no correlative, spatial information for the localization of specific, differentially truncated full length Aβ isoforms across and within single plaques can be retrieved as antibodies cannot directly distinguish entire peptide sequences, which in turn limits the biological information retrievable. In contrast, mass spectrometry (MS) methods in turn provide the chemical specificity necessary though have usually been used for ex-situ characterization of Aβ without spatial information[33–35].

The advent of mass spectrometry-based imaging (MSI) allowed to overcome the limitations of conventional pathological imaging techniques and allows for characterization of comprehensive metabolite, lipid and peptide patterns in situ at 5–10 µm resolution[36]. In the context of AD, MSI has been successfully demonstrated for mapping lipid and Aβ patterns at the single plaque level in both mouse models[37–40] and postmortem human brain tissue[41]. Further, morphologically heterogeneous amyloid pathology can be interrogated using advanced chemical microscopy with structure sensitive fluorescent probes, such as luminescent conjugated oligothiophenes (LCO) that provide both a qualitative and quantitative readout on amyloid cross β sheet content and on plaque morphology, respectively in an unbiased way[42–45]. Investigating high-content imaging data is however, challenging, time consuming and prone to bias. Deep learning has emerged as a transformative approach in the field of histopathology and MSI, enabling advanced analysis and interpretation of complex biological samples[46,47]. Recent studies have highlighted the efficacy of deep learning techniques in enhancing image classification accuracy and facilitating the identification of biomarkers in histological contexts[48,49].

In this work, we made use of those developments and devised an integrative, multimodal imaging paradigm based on correlative MALDI MSI and LCO microscopy to characterize heterogeneous plaque morphology and Aβ plaque pathology across different forms of AD amyloidosis (Fig. 1a, Tab. 2). We developed a deep learning-based method to identify plaque morphotypes in the hyperspectral LCO microscopy data space to subsequently guide correlative MS imaging experiments performed on the same tissue section. This approach identified specific Aβ patterns within different polymorphic plaque entities across different etiological subtypes of AD, including sporadic and familial forms of AD as well as cognitively unimpaired amyloid positive individuals with advanced pathology. This delineated and further characterized a previously reported coarse-grained plaque type[50] that represents a potential key phenotype of amyloid toxicity.

## Results
### Correlative chemical imaging and deep learning identify polymorphic plaque phenotypes across different clinical pathologies

Herein we set out to delineate amyloid peptide signatures of polymorphic plaque type across sporadic and familial forms of AD as well as in cognitively unimpaired amyloid positive individuals as controls with advanced pathology (Tab.2) to identify differences in plaque biochemistry related to clinical symptoms. For this we implemented a multimodal chemical imaging strategy based on MALDI MSI and fluorescent microscopy using functional amyloid probes (LCO) (Fig. 1a)[41]. Bisecting k-means clustering analysis (bkm-CA) based segmentation of the MALDI MSI data revealed characteristic chemically diverse localization patterns (pseudoclusters) that resemble plaque features across the tissue sections and are characterized by a specific chemical makeup. Alignment of the bkm-CA segmentation map with the LCO amyloid microscopy allowed for tentative identification of different plaque morphotypes that co-localized with distinct MSI segmentation derived plaque features (Fig.1b). In both CUAP and sAD, this LCO/MSI approach identified three clusters of plaque types, including diffuse plaques (DP) and dense-core plaques (CP) as histologically validated through chromogenic and fluorescent immunohistochemistry (Fig.1d,e) on independent tissue sections from one of the patients included in the present cohort. In fAD tissue, two main clusters were identified comprising only dense core plaques (CP) and for one patient a diffuse plaque type annotated as cotton wool plaque (CWP) (Fig. 1b)[51]. Most interestingly, in both sAD and fAD but not CUAP (Fig. 1c), the multivariate cluster analysis image analysis identified a characteristic subtype of dense-core plaques that showed a morphological distinct phenotype reminiscent of what has

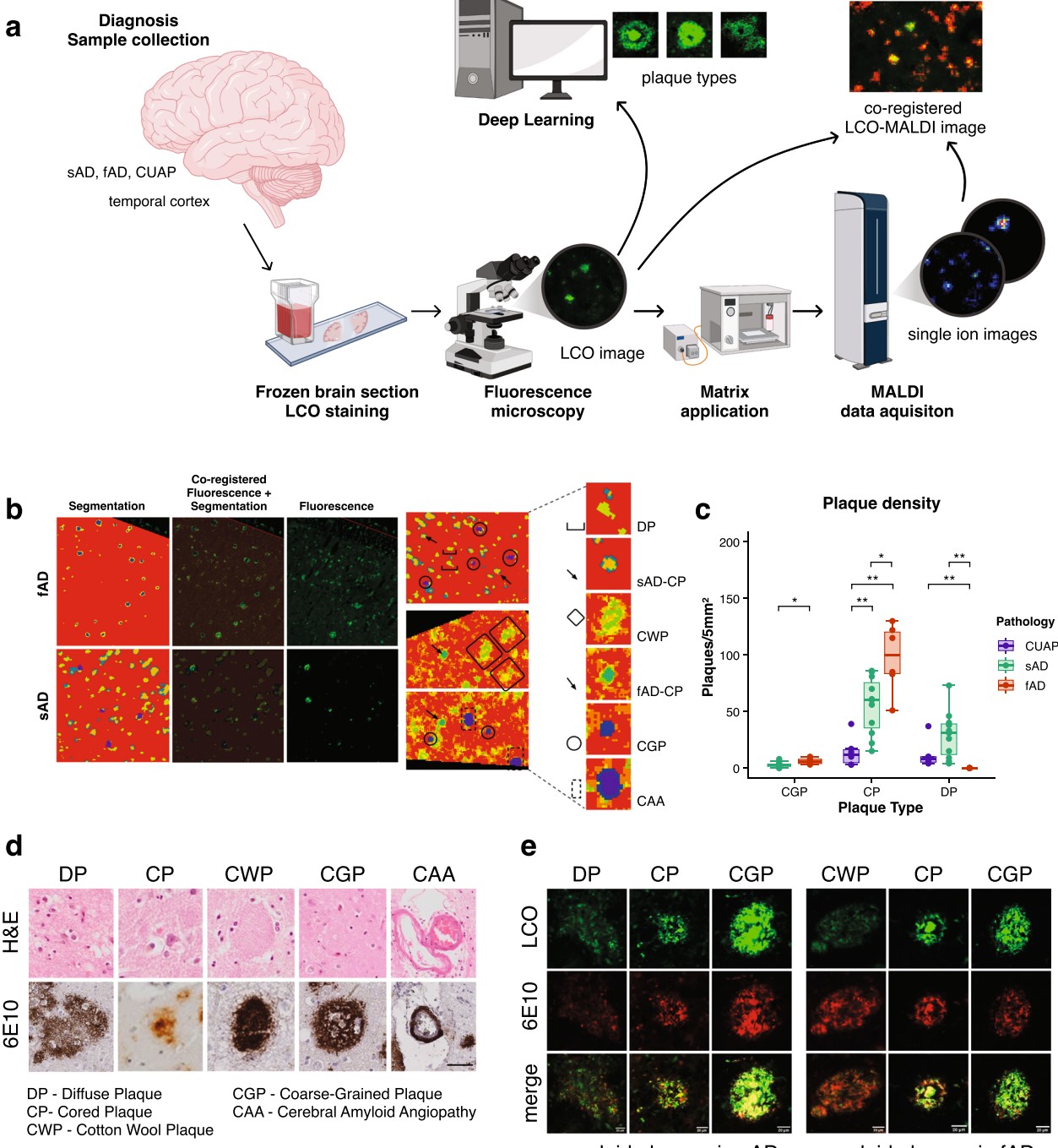

**Fig. 1 | Correlative chemical imaging of plaque polymorphism in Alzheimer's disease brain. a** Experimental strategy: Brain sections were analyzed with both MALDI MSI (for amyloid peptide imaging) and fluorescence microscopy using luminescent amyloid probes. The data analysis strategy included MSI spectra analysis, image correlation, segmentation and deep learning. Created in BioRender. Szadziewska, A. (2025) https://BioRender.com/q15x632. **b** Initial plaque analysis through registration of MALDI MSI segmentation maps with fluorescent amyloid staining (luminescent conjugated oligothiophenes, LCO). Bisecting k-means cluster analysis yielded spatial segmentation of MALDI MS imaging data and allowed to annotate different plaque types in brain tissue based on their chemical signatures encoded in pseudoclusters outlined as pixel by pixel pseudo-colored maps (e.g., red, yellow, green, blue). Different plaques identified in LCO guided MSI segmentation included diffuse plaques (DP), cored plaques (CP), cotton wool plaques (CWP), coarse-grained plaques (CGP) and CAA. **c** Plaque loads for all plaque types were determined across cognitively unimpaired amyloid positive (CUAP), sporadic Alzheimer's disease (sAD) and familial Alzheimer's disease (fAD) brain tissues.

Boxplots show the median and interquartile range (IQR, 25th–75th percentile), with whiskers extending to the smallest and largest values within 1.5 × IQR, sAD $n = 12$, fAD $n = 6$, CUAP n = 5, statistical analysis: Wilcoxon rank-sum test for: sAD-CGP vs. fAD-CGP $p = 0.034$, CUAP-CP vs sAD-CP $p = 0.009$, sAD-CP vs fAD-CP $p = 0.034$, CUAP-CP vs fAD-CP $p = 0.009$, CUAP-DP vs sAD-DP $p = 0.097$, CUAP-DP vs fAD-DP $p = 0.009$, sAD-DP vs fAD-DP $p = 0.006$. The resulting $p$-values were corrected for multiple comparisons by two-sided Benjamini-Hochberg method. *$p < 0.05$, **$p < 0.01$, ***$p < 0.001$. **d** Representative examples of diffuse plaque, cored plaque, cotton wool plaque, coarse-grained plaque and cerebral amyloid angiopathy through H&E and antiAβ (6E10) staining acquired from an independent section of the same patient. Scale bar: 40 μm. **e** Fluorescent microscopy of heterogenous plaque pathology. Representative examples from an independent tissue section showing diffuse plaque, cored plaque and coarse-grained plaque in sAD and cotton wool plaque, cored plaque and coarse-grained plaque in fAD through LCO/ antiAβ (6E10) co-staining. Scale bar: 20 μm.

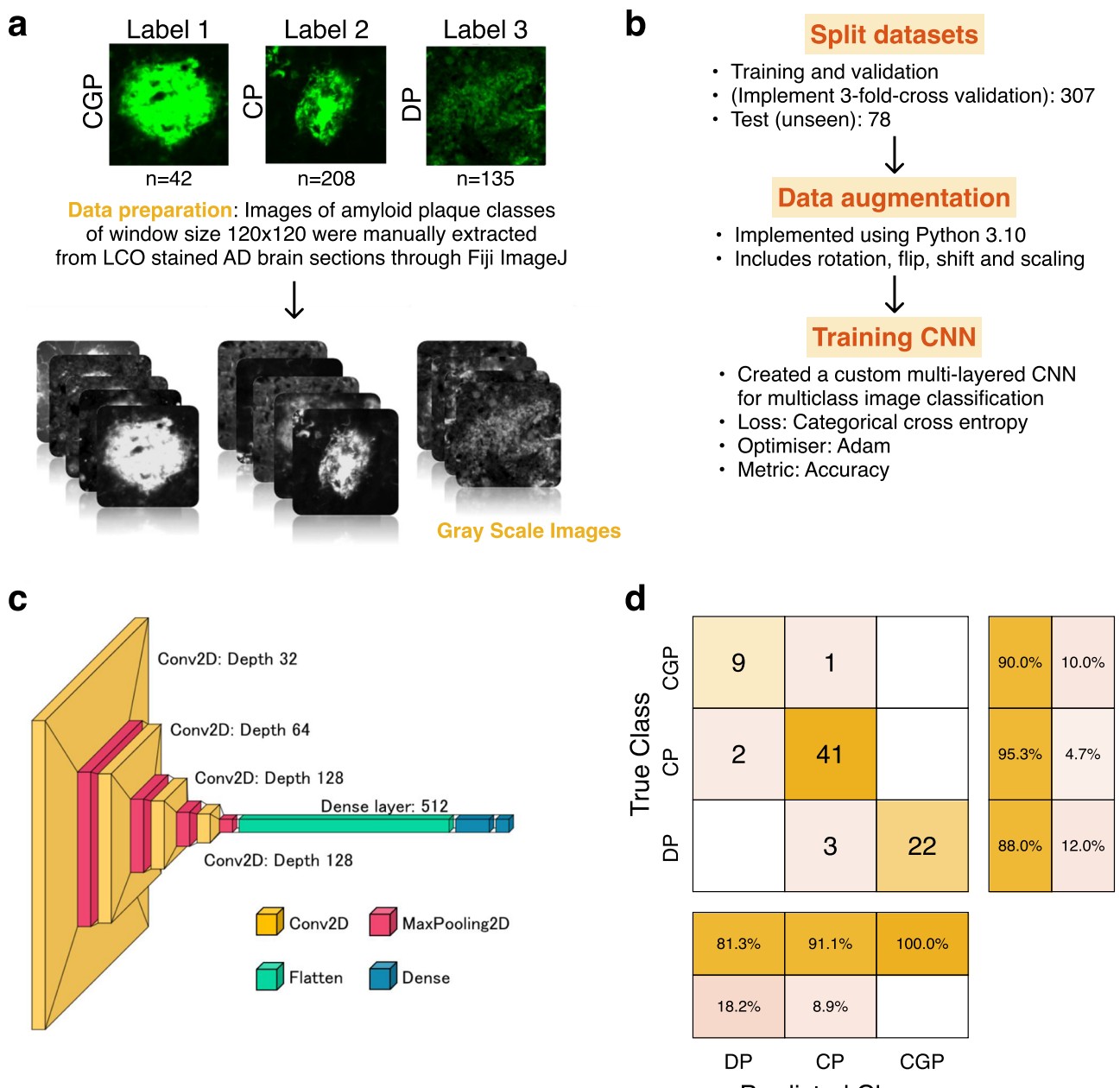

**Fig. 2 | Plaque classification using deep learning of functional amyloid microscopy data. a** Data preparation and preprocessing. Labelled images were manually extracted, categorised, and resized to a standard size (120 × 120). The images were converted to grayscale and adjusted for contrast. (CGP coarse-grained plaques, CP cored plaques, DP diffuse plaques) **b** Model training and development. The datasets were split into training (*n* = 307) and test sets (*n* = 78). Training datasets were split further for training and validation to execute three-fold cross validation and models were trained using various training parameters. The final model was obtaining by averaging the models from three-fold cross validation **c** Network architecture of the deep learning training model. **d** Results of the model evaluation using standard classification metrics. The final model achieved an accuracy of 92.3% in the unseen test set (*n* = 78).

previously been described as coarse-grained plaques[50] (Fig. 1c–e). Plaque loads determined for each patient showed that diffuse plaques were prominent in both CUAP and sAD. Cored plaques were observed across all CUAP, sAD and fAD, with increasing higher abundances from CUAP over sAD to fAD. Coarse grained plaques were only observed in AD, where the abundance was higher in fAD compared to sAD (Fig. 1c). No difference in coarse grained plaque load was observed between different APOE E3 and E4 allele carriers both in sAD and fAD.

Co-registration of LCO images onto MALDI MSI data for plaque classification is though subject to bias. However, this approach can serve as a valid method to investigate whether the plaque specific chemical signature encoded in the respective MSI data allow to differentiate the various plaque morphotypes. To address this issue of bias, we used an artificial intelligence (AI) strategy in developing a deep learning model to classify these plaques automatically and guide targeted plaque analysis. Employing the generated deep learning model onto the LCO tissue images allowed to guide subsequent MALDI MSI acquisition and ROI annotation based on the deep learning derived plaque classifications. In the confusion matrix, with a total of 78 instances, the model correctly classified 72 instances correctly indicating a strong ability to distinguish between the three classes (Fig. 2). However, the model also misclassified 6 instances suggesting some level of error in the classification into all three classes. Despite these misclassifications, the overall accuracy remained high at 92.3%,

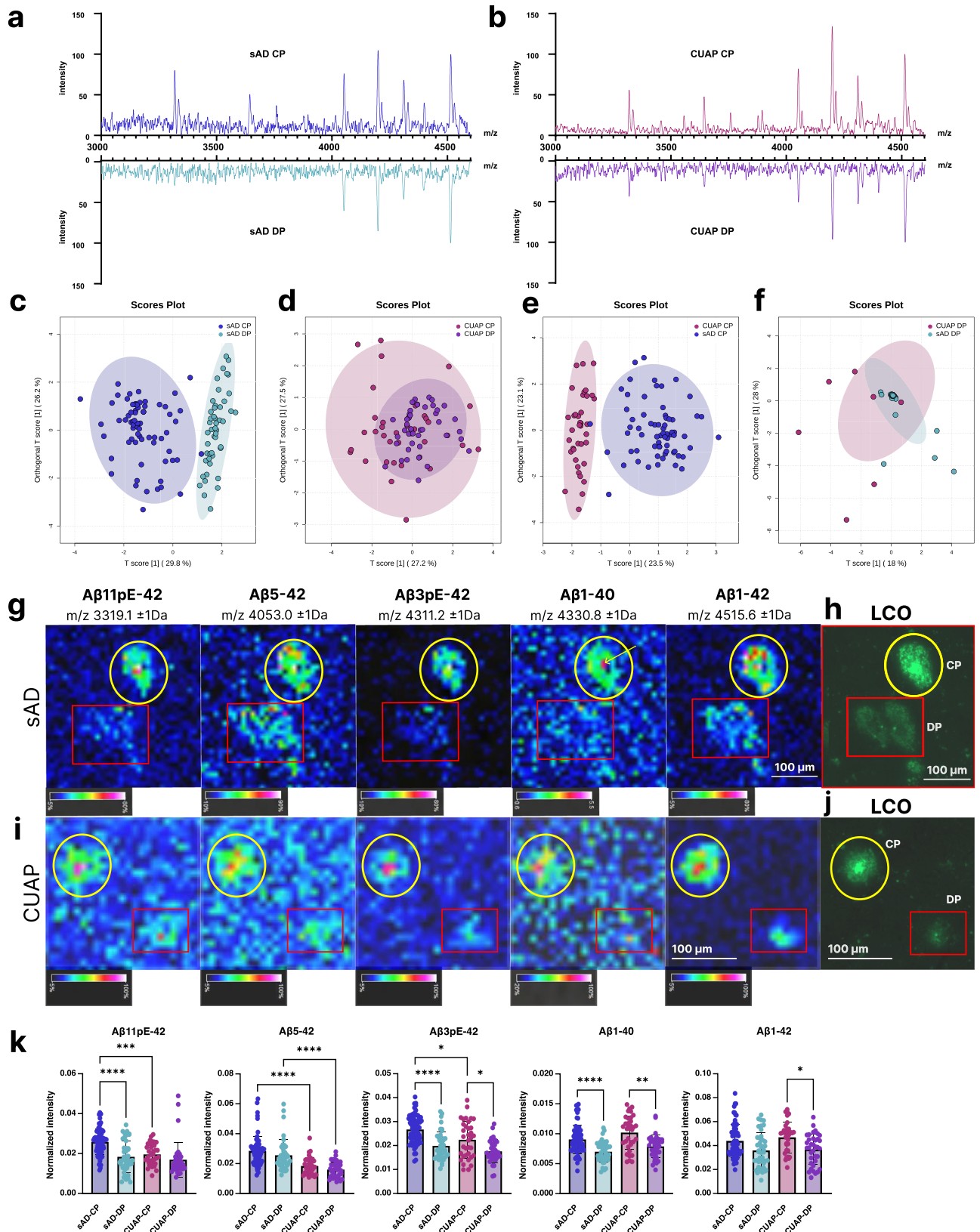

reflecting the model's proficiency in classifying instances correctly. Additionally, the recall of 0.90 (coarse-grained), 0.95 (cored), 0.88 (diffuse) suggests that the model effectively captured most true positive instances and precision values of 0.82 (coarse-grained), 0.91 (cored), and 1 (diffuse), which indicates a low rate of false positive predictions. The F1 score provided a balance between precision and

recall and is useful when the classes are imbalanced, as it was for this dataset. An F1 score (0.86: coarse-grained, 0.93: cored, 0.94: diffuse) closer to 1 indicates perfect precision and recall, while a score closer to 0 reflects poor performance. Our CNN-based approach achieved state-of-the-art performance in classifying amyloid plaque images into coarse-grained, cored, and diffuse categories. The model

**Fig. 3 | Diffuse and cored plaques in sporadic AD and cognitively unimpaired amyloid positive. a**, **b** Representative average mass spectra obtained for cored (CP) and diffuse plaques (DP) from all ROI/plaque type in one sporadic Alzheimer's disease (sAD) and one cognitively unimpaired amyloid positive (CUAP) patient. **c**–**f** OPLS-DA score plots comparing diffuse and cored plaques in **c** sAD and (**d**) CUAP. **e** OPLS-DA score plots for comparing cored plaques in between sAD and CUAP. **f** OPLS-DA score plots for diffuse plaques across sAD and CUAP **g**–**i** Representative MALDI MSI single ion images (**g**, **i**) and correlative LCO microscopy (**h**, **j**) of statistically significant Aβ peptides obtained from one patient per group (sAD, CUAP). Intensity scale: rel. intensity in %. Scalebar: 100 μm.

**k** Corresponding bar plots showing univariate comparison of single ion intensities with respective plaque types. Plots indicate single plaque values mean±SD. Number of patients: $n = 12$ sAD and $n = 5$ CUAP; number of plaques per patient and plaque type $N = 5$–15. Univariate analysis was performed using Kruskall-Wallis test with two-sided Dunn's correction: Aβ11pE-42: sAD-CP vs. sAD-DP $p < 0.0001$, sAD-CP vs. CUAP-CP $p = 0.0008$; Aβ5-42: sAD-CP vs. CUAP-CP $p < 0.0001$, sAD-DP vs. CUAP-DP $p < 0.0001$; Aβ3pE-42: sAD-CP vs. sAD-DP $p < 0.0001$, sAD-CP vs. CUAP-CP $p = 0.0185$, CUAP-CP vs. CUAP-DP $p = 0.0232$; Aβ1-40: sAD-CP vs. sAD-DP $p < 0.0001$, CUAP-CP vs. CUAP-DP $p < 0.01$; Aβ1-42: CUAP-CP vs. CUAP-DP $P < 0.05$ (*$p < 0.05$, **$p < 0.01$, ***$p < 0.001$, ****$p < 0.0001$).

demonstrated high accuracy and robustness across diverse plaque morphologies. Furthermore, qualitative analysis revealed the model's capability to capture intricate features associated with each plaque type.

## Amyloid fibrillization is associated with increased Aβx-42 and Aβ1-40 deposition

Following the deep learning based classification, the aim was then to determine the Aβ signatures associated with different plaque morphologies in brain tissue across different AD etiologies. In AD brain tissue, Aβ plaques comprise both diffuse- and cored deposits, including neuritic, cored plaques (NP) whose abundance has been linked to cognitive symptoms and disease progression[52]. In contrast, in CUAP, predominantly diffuse plaques were observed along with some cored plaques but generally no neuritic plaques (Supplementary Fig. 1)[28,52–54]

This suggests that increased formation of cored plaques with age and most importantly neuritic, cored plaques is critically associated with the development cognitive symptoms and AD respectively and that the molecular basis for plaque maturation is reflected in their Aβ signature.

Here, LCO and deep learning identified plaque type specific regions of interest (ROI) (Fig. 1 and Supplementary Fig. 1) and guided subsequent MSI acquisition. To quantify the Aβ profile of single plaques morphotypes, ROI mass spectrum data were extracted for diffuse- and cored plaque ROI in both CUAP and sAD (Fig. 3a, b) and compared by means of multivariate analysis (OPLS-DA) and follow up univariate group statistics. Here analysis was performed for a consistent set of Aβ peptides that were detected in all plaques across all the groups that were compared.

First, OPLS-DA based multivariate analysis of the plaque spectral data separated cored and diffuse plaque types in sAD based on their Aβ signatures (Fig. 3c, d). The primary loadings that underlie those separations (Variable importance in Projection, VIP) reveal the Aβ signature associated with the respective plaque types (Supplementary Fig. 2a–d). Here, the OPLS-DA derived VIP scores showed Aβ1-40 and Aβ1-42 along with N-terminally truncated, pyroglutamated forms (pE) of Aβx-42, i.e., Aβ3pE-42, Aβ11pE-42 as the critical Aβ species discriminating diffuse plaques from cored plaques in sporadic AD (Supplementary Fig. 2a). Univariate analysis of individual plaque data confirmed significantly higher levels of Aβ11pE-42, Aβ3pE-42 and Aβ1-40 in cored plaques as compared to diffuse plaques as further illustrated by representative single ion maps (Fig. 3g, k). For CUAP patients no clear separation was observed for cored and diffuse plaques, though the univariate analysis revealed higher levels in Aβ3pE-42, Aβ1-40 and Aβ1-42 in cored compared to diffuse plaques like in sAD though less pronounced (Fig. 3b, d, i, k and Supplementary Fig. 2b). Clustered heat maps of averaged peptide intensities further support the OPLS-DA patterns showing a distinct separation of cored- and diffuse plaques in sAD (Fig. 3a, Supplementary Fig. 3a, b) that however was not prominent on the multivariate level for CUAP (Fig. 3d, Supplementary Fig. 3b). Similarly, cored plaques in sAD were distinctively separated from cored plaques in CUAP (Fig. 3e, Supplementary Fig. 3c), while diffuse plaques across sAD and CUAP were not separated (Fig. 3f, Supplementary Fig. 3d).

For cored plaques in sAD, colocalization analysis showed that the primary site of Aβ1-40 deposition was the core, while Aβ1-42 and Aβ3pE-42 deposition was homogeneous across the entire plaque area (Fig. 3g, h, Supplementary Fig. 1b).

This suggests that cored plaque formation in both sporadic AD and CUAP is predominantly associated with deposition of Aβ1-40 at the core as well as increased deposition of Aβ1-42, Aβ3pE-42 and Aβ11pE-42, which is more pronounced for cored plaques in AD than in CUAP (Fig. 3k). These results along with a larger number of diffuse plaques in CUAP (Fig. 1c) suggests that diffuse plaques might represent immature precursors of cored plaques and that maturation of diffuse plaques into cored plaques and eventually into neuritic phenotypes is a process associated with developing AD[29].

In fAD, non-fibrillar/diffuse plaques were observed only within one fAD individual (with a double substitution in *PSEN1*; A434T & T291A)[55] showing a morphological phenotype attributed as cotton wool plaques (CWP). This plaque type could be delineated based on its MSI profile and validated by correlative LCO microscopy (Fig S3a, b). OPLS-DA based multivariate analysis showed a separation for the cotton wool- and cored plaque populations. VIP scores suggest Aβ3pE-42 as the critical peptide discriminating cotton wool from cored plaques in fAD (Fig. S3c, d). The MSI patterns of cotton wool plaques were similar to cored plaques in fAD, showing prominent levels of Aβ11pE-42, Aβ3pE-42, Aβ2-42, Aβ4-42 and Aβ1-42 along with some Aβ1-40 content. However, the relative abundance of Aβ11pE-42/ Aβ1-42, Aβ11-42/ Aβ1-42 and Aβ3pE-42/ Aβ1-42 was characteristically higher in cotton wool plaques than in cored plaques (Supplementary Fig. 4e).

## Dense-core plaques in sAD and fAD brain show different content of axonal dystrophy

Following the initial analysis on cored and diffuse plaques, we then investigated the dense-core plaque phenotypes observed both in sAD and fAD. Interestingly, our cluster analysis based image segmentation of the LCO/MSI data revealed two subtypes of dense-core plaques annotated as (classic) cored plaques and coarse-grained plaques (Fig. 1b–e) which is in line with recent data[50]. Using immunohistochemical and morphological characterization, we investigated the relation of these different plaques with paired-helical filament (PHF-1) Tau positive dystrophic neurites as indicators for neurotoxicity. Cored-, coarse-grained, diffuse- and cotton wool plaques were visualized with LCO probes that allow to stain both mature and prefibrillar amyloid aggregates and outline the characteristic plaque morphologies. The LCO microscopy was multiplexed with immune-fluorescent labelling towards PHF Tau and reticulon-3 (RTN3) to identify the presence of Tau filament positive neurites (PHF-1) and dystrophic neurites (RTN3), respectively, within different plaque morphotypes (Fig. 4, Table 2) In sAD, coarse-grained plaques showed a more prominent staining for PHF-1 as compared to cored plaques, while diffuse plaques showed no PHF-1/RTN3 staining. Similarly, to PHF-1, RTN3 showed increased staining in coarse-grained plaques compared to cored plaques (Fig. 4a). In fAD brain tissue, the most prominently cored plaque and coarse-grained plaque pathology was observed in all patient samples along with cotton

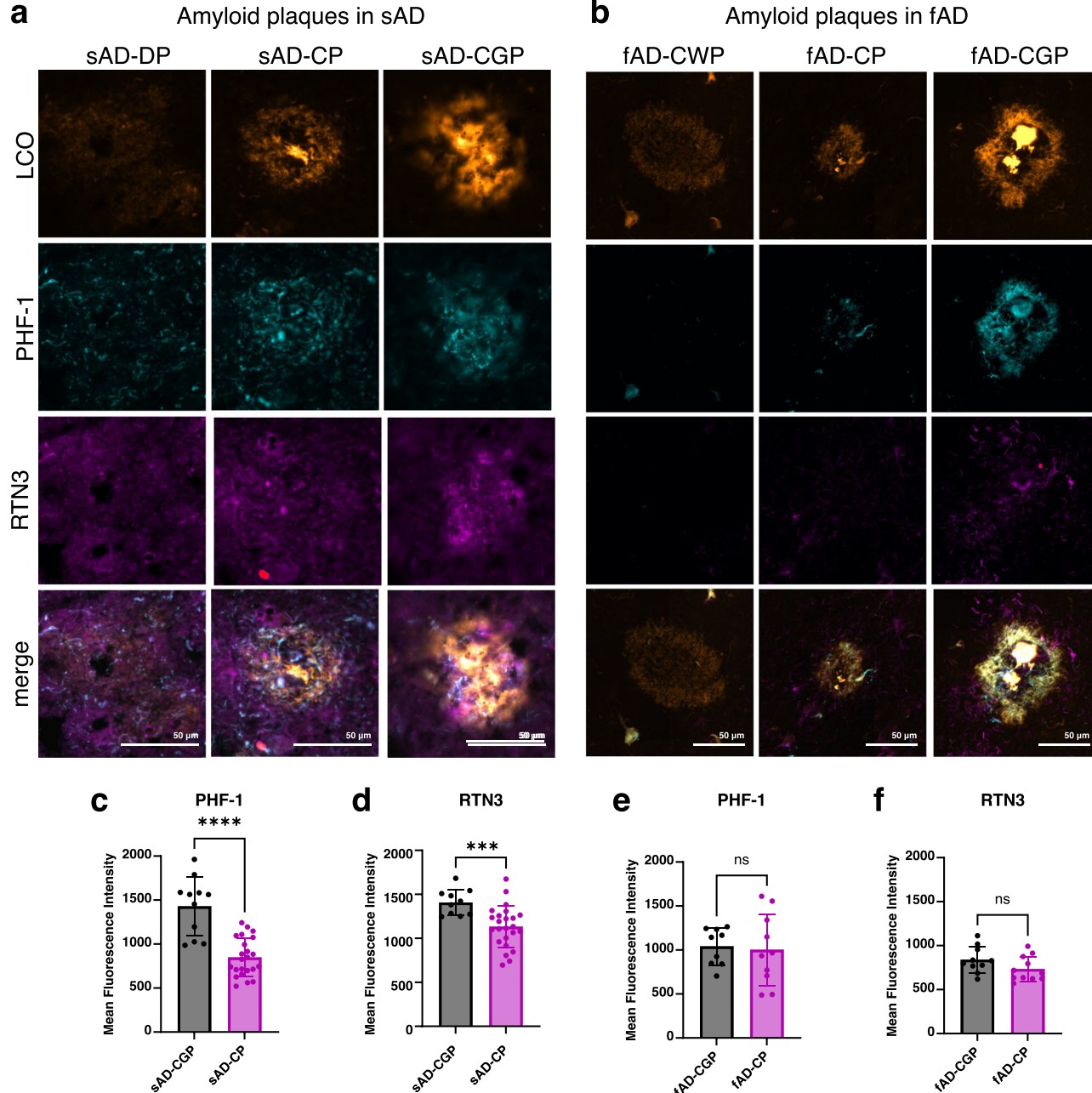

**Fig. 4 | Quantification of neurites in Aβ plaque pathology. a** Diffuse (DP), cored (CP) and coarse-grained plaques (CGP) in sporadic Alzheimer's disease (sAD) visualized by LCO based amyloid staining. Localization of PHF antibody (parahelical filament, PHF-1) staining of the same plaques indicates almost no to very feeble colocalization in diffuse and in cored plaques while a prominent colocalization of LCO with PHF-1 is observed within coarse-grained plaques. Similarly, RTN3 (Reticulon 3) indicates dystrophic neurites. **b** Cotton wool (CWP), cored (CP) and coarse-grained plaques (CGP) in familial Alzheimer's (fAD) disease visualized by LCO staining. Localization of the PHF-1 staining of the same plaques indicates almost no to very feeble colocalization in cotton wool plaques and in cored plaques while a prominent colocalization of LCO with PHF-1 is observed within coarse-grained plaques. Similarly, RTN3 indicates dystrophic neurites. **c**–**f** Boxplots (Mean ±SD) indicating mean fluorescence for the entire plaque for **c** PHF-1 in sAD-CG and sAD-CP, (**d**) RTN3 in sAD-CG and sAD-CP, (**e**) PHF-1 in fAD-CG and fAD-CP, (**f**) RTN3 in fAD-CG and fAD-CP. Number of patients sAD: *n* = 3 and fAD: *n* = 2; number of plaques: sAD: *N* = 11 CGP and 21CP; fAD: *N* = 9 CGP and *N* = 11 CP. Univariate analysis was performed using two-sided Mann-Whitney U test: PHF-1: sAD-CG vs. sAD-CP, *p* < 0.0001, RTN3: sAD-CG vs. sAD-CP, *p* = 0.0006 (***$p < 0.001$, ****$p < 0.0001$).

wool plaques for one patient (Supplementary Fig.1b). No PHF-1 or RTN3-positive neurites were observed in cotton wool plaques (Fig. 4b). In contrast, both cored and coarse-grained plaques in fAD showed positive staining for PHF-1 and RTN3 dystrophic neurites. To quantify levels of neuritic content in a plaque type specific manner, we measured the mean fluorescence intensity of RTN3 and PHF-1 of LCO positive stained plaque area in sAD and fAD cored and coarse-grained plaques. The results showed that in sAD, the fluorescence

intensity of PHF-1 (Fig. 4a, c) and RTN3 (Fig. 4a, d) is higher in coarse-grained plaques than cored plaques. In contrast, both plaque types show equal levels of neuritic staing in fAD (Fig. 4b, e, f). Of note, we observed a higher plaque load of coarse-grained plaques in fAD that also show earlier onset of disease. Similar to Boon et al.[50] who found that coarse-grained plaque loads were higher in EOAD compared to LOAD, this suggests that coarse-grained plaques are associated with disease onset and progression (Fig.1c).

**Table 2 | Patient Demographics**

| ID | AAD | Sex | Path Diagnosis | APOE | Braak stage | Thal stage | CERAD | ABC | Mutation |
|---|---|---|---|---|---|---|---|---|---|
| sAD1 | 56-60 | F | AD | 33 | VI | 5 | 2 | A3B3C2 | *NA* |
| sAD2 | 63-67 | M | AD | 44 | VI | 5 | 3 | A3B3C3 | *NA* |
| sAD3 | 67-69 | M | AD | 34 | VI | 5 | 3 | A3B3C3 | *NA* |
| sAD4 | 64-68 | F | AD | 33 | VI | 5 | 3 | A3B3C3 | *NA* |
| sAD5 | 77-80 | M | AD | 33 | VI | 5 | 3 | A3B3C3 | *NA* |
| sAD6 | 68-70 | F | AD | 34 | VI | 5 | 2 | A3B3C2 | *NA* |
| sAD7 | 62-64 | M | AD | 33 | VI | 5 | 3 | A3B3C3 | *NA* |
| sAD8 | 63-65 | F | AD | 34 | VI | 5 | 3 | A3B3C3 | *NA* |
| sAD9 | 65-67 | F | AD | 33 | VI | 5 | 3 | A3B3C3 | *NA* |
| sAD10 | 68-70 | F | AD | 44 | VI | 5 | 2 | A3B3C2 | *NA* |
| sAD11 | 73-75 | M | AD | 44 | VI | 5 | 3 | A3B3C3 | *NA* |
| sAD12 | 85-88 | F | AD | 34 | VI | 5 | 3 | A3B3C3 | *NA* |
| fAD1 | 45-48 | M | FAD | 33 | V | 5 | 3 | A3B3C3 | *PSEN1 A434T & T291A* |
| fAD2 | 64-67 | F | FAD | 34 | VI | 5 | 3 | A3B3C3 | *PSEN1 after 200 (R278I)* |
| fAD3 | 50-52 | F | FAD | 44 | VI | 5 | 3 | A3B3C3 | *PSEN1 Intron 4* |
| fAD4 | 36-39 | F | FAD | 33 | VI | 5 | 3 | A3B3C3 | *PSEN1 E120K* |
| fAD5 | 57-60 | F | FAD | 34 | VI | 5 | 3 | A3B3C3 | *PSEN1 E184D* |
| fAD6 | 49-52 | M | FAD | 33 | VI | 5 | 3 | A3B3C3 | *PSEN1 mutation (Intron 4)* |
| CUAP1 | 90-92 | F | Pathological ageing | 33 | IV | 5 | 2 | A3B2C2 | *NA* |
| CUAP2 | 85-88 | F | Pathological ageing | 34 | IV | 4 | 2 | A3B2C2 | *NA* |
| CUAP3 | 85-87 | F | Pathological ageing | 34 | I | 4 | 1 | A3B1C1 | *NA* |
| CUAP4 | 102-104 | F | Pathological ageing | 23 | IV | 5 | 1 | A3B2C1 | *NA* |
| CUAP5 | 90-92 | F | Pathological ageing | 34 | IV | 5 | 2 | A3B2C2 | *NA* |

*AAD* age at death, *CUAP* Cognitively Unimpaired Amyloid Positive, *PCA* posterior cortical atrophy, *PPA* primary progressive aphasia, *APOE* apolipoprotein E genotype, *CERAD* Consortium to Establish a Registry for Alzheimer's Disease, *ABC* Combined score based on Thal phase, Braak stage and CERAD Score, *PSEN1* Presenilin 1.

## Coarse-grained plaques are characterized by high Aβx-40 deposition

Following the immunohistochemical and morphological characterization of cored- and coarse-grained plaques we then aimed to interrogate the Aβ profiles associated with these plaque populations through MALDI MSI. The distinct Aβ patterns allowed us to identify the plaque types through clustering-based image segmentation (Fig. 1) and train a deep learning model for unbiased detection in large tissue areas that ultimately guided the subsequent MALDI MSI analysis.

Inspection of the plaque ROI spectral data showed prominent differences in Aβ pattern across cored and coarse-grained plaques, particularly with regard to the major species Aβ1-40 and Aβ1-42 (Fig. 5a, b). OPLS-DA showed a clear separation of CP and CGP in both sAD (Fig. 5c, Supplementary Fig. 5a, b) and in fAD (Fig. 5d, Supplementary Fig. 5c, d). In contrast, no clear separation was observed in between cored plaques across sAD and fAD (Fig. 5e, Supplementary Fig. 5e, f) as well as for coarse-grained plaques between fAD and sAD (Fig. 5f, Supplementary Fig. 5g, h).

The VIP of the OPLS-DA and subsequent univariate statistics revealed that coarse-grained plaques show higher levels of Aβx-40 species including Aβ4-40, Aβ3pE-40 and Aβ1-40 as compared to cored plaques with both sAD and fAD, as prominently discernable from the respective MALDI MSI single ion images (Fig. 5g). In contrast, both in sAD and fAD, Aβx-42 species were higher in cored plaques as compared to coarse-grained plaques including Aβ11pE-42, Aβ5-42, Aβ3pE-42 and Aβ1-42 (Fig. 5g, h).

Comparison of cored plaques across sAD and fAD, showed higher levels of Aβ5-42 and Aβ1-42 in sAD cored plaques as compared to cored plaques in fAD (Fig. 5h). No difference was observed for the other Aβ species in cored plaques (Fig. 5h). OPLS-DA and univariate analysis

found no statistical difference in between coarse-grained plaques from sAD and fAD patients.

Clustered heat maps of averaged peptide intensities showed a similar pattern as observed in OPLS-DA showing a distinct separation of cored- and coarse-grained plaques in both sAD and fAD (Supplementary Fig. 5i, j). In contrast, cored plaques in between sAD and fAD as well as coarse grained plaques across sAD and fAD were not separated (Supplementary Fig. 5k, l). Correlations matrices generated for cored plaques in sAD (Supplementary Fig. 6) and fAD (Supplementary Fig. 7) further confirmed their relative similarity, showing similar correlation patterns within the Aβ isoforms. The regression analysis data showed positive correlation of different Aβx-42 truncations with each other, while Aβx-40 peptides display positive correlation with other Aβx-40 species (Supplementary Figs. 6, 7). Similarly, regression analyses of peptide species in coarse-grained plaques showed very similar patterns for both sAD and fAD plaques (Supplementary Figs. 8, 9). The results showed a significant, negative correlation of Aβ1-40, Aβ3pE-40, Aβ3-40 with Aβx-42 species. In contrast, many N-terminal Aβx-42 truncations showed a positive correlation with other Aβx-42 species (Supplementary Figs. 8, 9). These data further confirm that coarse-grained plaques are very similar across AD etiologies.

## Coarse-grained plaques display amyloid signatures designative to CAA

The MS signatures observed for coarse-grained plaques were dominated by Aβ1-40 species resembling amyloid signatures obtained for CAA (Fig. 6a) and well in line with previously described IHC and MS data for CAA[25,33]. We therefore compared MALDI MSI Aβ patterns between CAA and coarse-grained plaques using OPLS-DA of the MSI derived ROI spectral data (Fig. 6b). Despite a weak separation

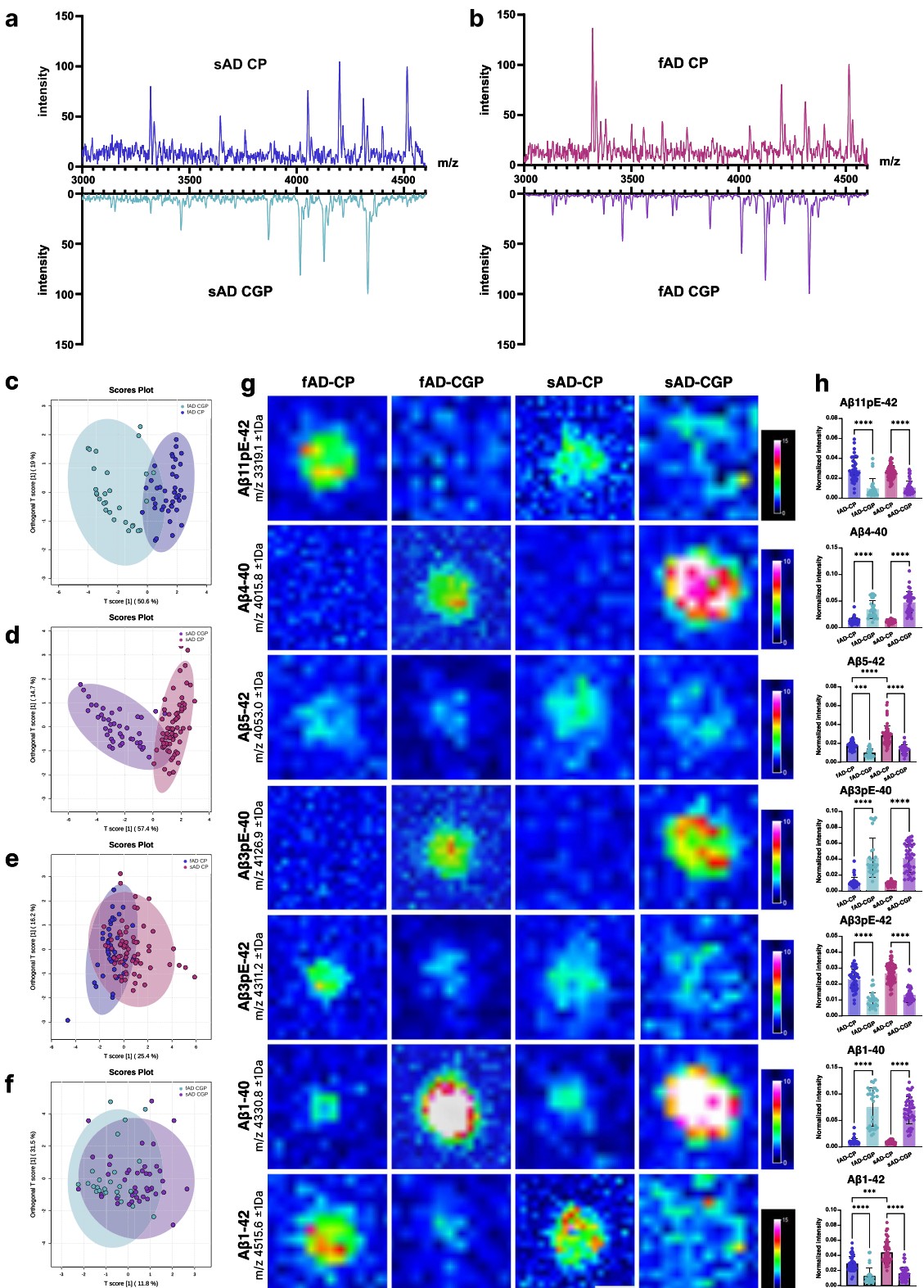

**Fig. 5 | Comparison of peptide patterns in dense-core plaques. a, b** Representative cored (CP) and coarse-grained plaque (CGP) specific average mass spectra obtained for multiple ROI/plaque type in one sporadic Alzheimer's disease (sAD) and one familial Alzheimer's disease (fAD) patient. **c, d** OPLS-DA score plots for comparison of CP and CGP spectral data across sAD (**c**) and fAD (**d**) showing separation of the populations. **e** OPLS-DA of CP in between sAD and fAD indicated partial separation of fAD CP and sAD CP, (**f**) while no separation was observed for CGP in between sAD and fAD. **g** Representative single ion images of statistically significant Aβ peptides obtained from one patient per group (sAD, fAD). Intensity scale: rel. intensity in %. Scalebar: 50 μm. **h** Corresponding bar plots showing univariate comparison of single ion intensities across respective plaque types. Plots indicate single plaque values mean±SD. Number of patients: $n = 12$ sAD and n = 6 fAD; number of plaques per patient and plaque type $N = 5–15$. Univariate analysis was performed using Kruskall-Wallis test with two-sided Dunn's correction: Aβ5-42 fAD-CP vs. fAD-CGP $p = 0.0002$; Aβ1-42 fAD-CP vs. sAD-CP $p = 0.0003$.; (***$p < 0.001$, ****$p < 0.0001$).

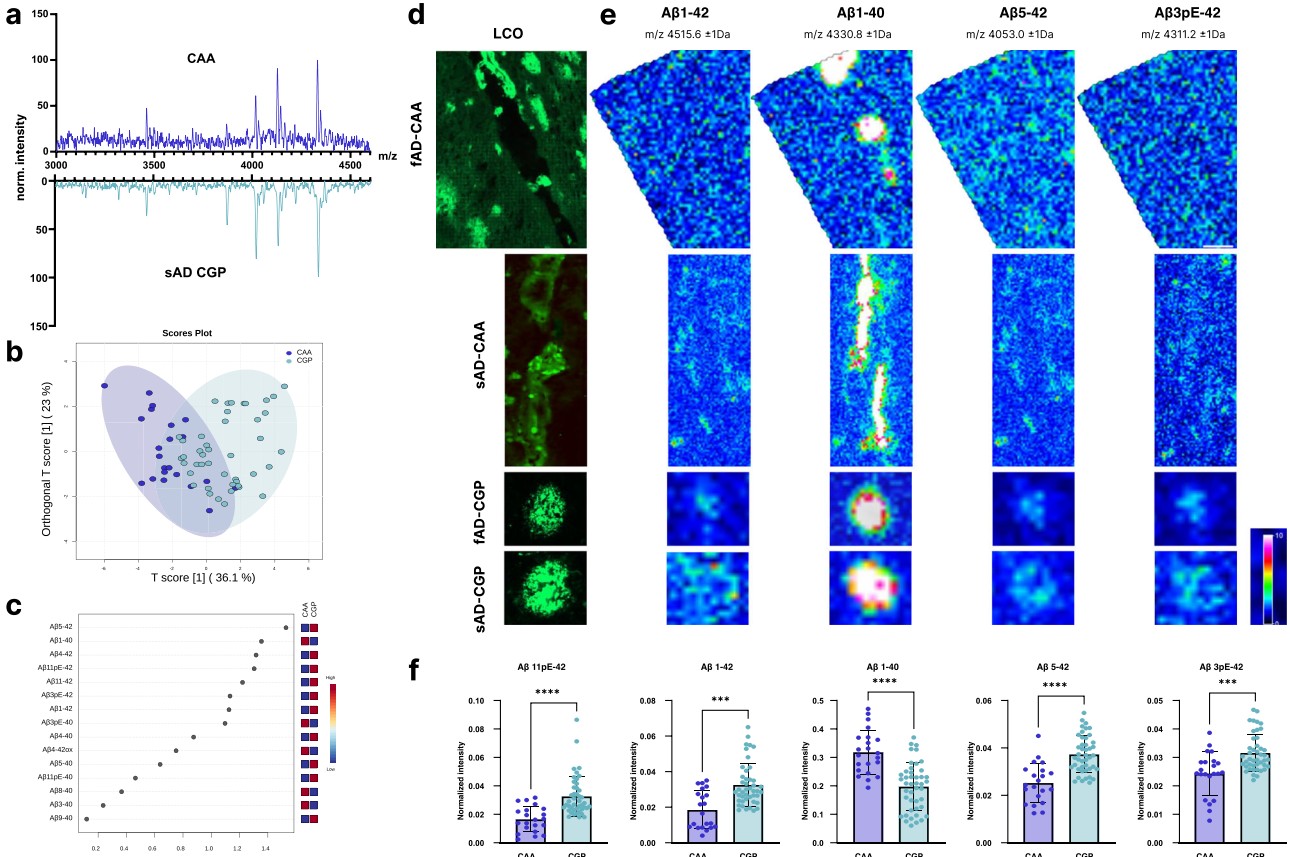

**Fig. 6 | Coarse-grained plaques display amyloid signatures designative to CAA.**
**a** Representative coarse-grained plaque (CGP) and cerebral amyloid angiopathy (CAA) specific average mass spectra obtained for several ROI/plaque type in one sporadic Alzheimer's disease (sAD) and one familial Alzheimer's disease patient (fAD). **b** OPLS-DA of CAA and CGP show partial separation (OPLS-DA model characteristics: R2X-0.329; R2Y-0.439; Q2-0.402). **c** Variable importance in Projection (VIP) scores for this OPLS-DA model. **d** LCO microscopy and correlative

**e** MALDI MSI single ion maps of different Aβ species. Intensity scale: absolute Intensity 1-10, Scalebar = 50 μm. **f** Univariate comparison between CAA and CGP. Plots indicate single plaque/CAA values (mean ± SD). Univariate analysis was performed using two tailed Mann-Whitney U test: Aβ11pE-42 $p < 0.0001$, Aβ1-42 $p = 0.0002$, Aβ1-40 $p < 0.0001$, Aβ5-42 $p < 0.0001$, Aβ3pE-42 $p = 0.0004$. *$p < 0.05$, **$p < 0.01$, ***$p < 0.001$, ****$p < 0.0001$. Number of patients $n = 10$ (sAD: 5; fAD:4); number of deposits $N = 21$ (CAA), $N = 46$ (CGP).

indicative of a large chemical overlap, the OPLS derived VIP and univariate statistical analysis showed lower levels of Aβ1-40 and Aβ3pE-40 in coarse-grained plaques as compared to CAA (Fig. 6c). Conversely, Aβx-42 species including Aβ1-42, Aβ5-42, Aβ3pE-42, and Aβ11pE-42 were increased in coarse-grained plaques as compared to CAA (Fig. 6c–f). The orthogonal pattern in more soluble Aβx-40 profiles for CAA and aggregation prone Aβx-42 content in coarse-grained plaque provide an explanation for their parenchymal deposition and retention.

## Discussion

The formation and evolution of structurally, heterogeneous (i.e., polymorphic) amyloid plaques have long been implicated in AD pathogenesis and progression[13,25,56]. Consequently, it has long been hypothesized that plaque polymorphism is associated with a distinct biochemical makeup, which however has been challenging to delineate with conventional biochemical staining techniques.

In this study, we developed a A.I. enhanced multimodal chemical imaging paradigm for unambiguous, comprehensive, spatial characterization of Aβ signatures of distinct plaque morphotypes across the Alzheimer's disease continuum at the single plaque level.

The main results of this work show that: (i) dense-core plaque formation is characterized by progressive deposition of Aβx-40 along with higher levels of AβxpE-42 as compared to diffuse plaques; ii) this difference is also observed in non-demented amyloid positive

individuals; (iii) a multi-cored plaque phenotype, previously annotated as coarse-grained plaque was detected and chemically characterized solely in AD and (iv) that coarse-grained plaques showed higher neuritic content and higher abundance in fAD patients, which had all an earlier disease onset compared to the sAD patients with later age of onset.

The molecular factors contributing to the formation of distinct Aβ plaque morphotypes[30,57] are not yet fully understood, despite their potential relevance for the pathogenesis of AD. Seminal work using histological and immunostaining techniques has established that diffuse plaques precede cored plaque formation[13,25,29,56] and that this process involves the deposition of N-terminal truncation[29,57,58].

Previous efforts to characterize plaques in situ relied mostly on chemical stains (e.g., Bielschowsky silver stain, Congo red, Thioflavin; for review see ref. 14) and immunohistochemistry towards Aβ[59]. Those experiments were complemented with ex situ characterizations using Western blot analysis providing additional, bottom-up data on plaque biochemistry[60] and in part at striking molecular weight resolution[61]. These conventional tools however have limitations with respect to molecular specificity and correlative spatial detection to fully characterize plaque pathology in situ across single plaque entities in an untargeted and unbiased way. The advent of biological mass spectrometry[33-35] and in part by combination with immunoprecipitation[62-64] greatly increased the resolution of those

experiments though was still limited to ex situ analysis of tissue extracts or laser microdissected plaques[41,65].

Herein we developed a multimodal chemical imaging approach to address this challenge. We employed functional amyloid microscopy based on LCO staining to unambiguously outline plaque morphotypes in situ in an unbiased way using a deep learning model. This approach then guided correlative mass spectrometry imaging experiments performed on the same tissue section.

The LCO method has previously identified plaque polymorphism across sAD and fAD pathology[43]. In combination with laser micro-dissection and immunoprecipitation, distinct peptide signatures could also be attributed to specific plaque morphotypes, though at the limitation of pooling hundreds of plaques[41]. The integration with MSI, as demonstrated here, facilitated the correlation of these orthogonal chemical signatures by refining Aβ signatures to structurally distinct plaque types in a comprehensive, untargeted and correlative way. The here presented deep learning approach minimized subjective bias and did allow to overcome issues with potential mis-annotations along with significantly streamlining the subsequent, correlative MSI acquisition. Here, the specificity of MS allowed to fully characterize different, full length Aβ isoforms in situ, thereby expanding the possibilities of previously used methods for plaque characterization.

A first aim was to investigate the differences of cored and diffuse plaque morphotypes in AD as well as in non-demented, amyloid positive individuals with advanced plaque pathology (CUAP, Thal 4). Our rationale for using Thal 4-staged CUAP was that this is a more appropriate comparison towards the symptomatic AD patients as differences in plaque chemistry can then be linked to clinical manifestation in this cross-sectional study, where no longitudinal data were available. We rationalized that if one were to investigate Thal 1 or 2 cases, one could not know where these cases would end up over time, i.e., whether they would end up being symptomatic AD or CUAP cases.

Here, a main finding using this chemical imaging approach was that cored plaques in both CUPA and sAD showed higher levels of Aβ1-40 along with marginally higher Aβ1-42 levels as well as more N-terminal truncation of Aβ1-42 including mainly Aβ3pE-42 and Aβ11pE-42. These findings are well in line with previous results using conventional immunodetection methods[25,29,30,57,66] and further support these pioneering studies. This is of interest as many of these previous observations were based on indirect measures such as in situ IHC and ex situ western blots or immunoprecipitation mass spectrometry (IP-MS). Herein we present ultimate proof of these earlier findings and refine those observations towards distinct plaque types for individual full length Aβ peptides.

Here, a particular question of interest relates to diffuse plaques and whether they represent premature, early precursors to classic senile plaques. Since the first description over 100 years ago[17] and their morphological characterization[15,67–69], diffuse plaques have been of great interest in studying the evolution in plaque pathogenesis. This is mainly related to findings showing their predominant abundance in non-demented amyloid positive[26–28,54,70], their lack of PHF-1 and Tau positive dystrophic neurites[70] and their early deposition in Down's syndrome[25] preceding cored plaques. Cumulatively, this suggested diffuse plaques to be immature, early amyloid aggregates in precipitating plaque pathology, particularly motivating to investigate their biochemical configuration in contrast to cored plaques. Here, in line with our results, previous studies, have shown that cored plaques are characterized by progressive Aβx-40 deposition[25,29,33,71] and relatively higher degree of N-terminal truncation, including mainly Aβ3pE-x and Aβ11pE-x pyroglutamation as compared to diffuse plaques[25,29,63,72]. From a mechanistic point of view, the deposition of Aβx-40 peptides upon cored plaque formation can be explained by concomitantly increased levels of Aβ3pE-42 and Aβ11pE-42 in cored plaques that facilitates the deposition of Aβx-40 peptides that are less prone to aggregation. Pyroglutamation represents an N-terminal functionalization of Aβ through enzymatic conversion of glutamate, which alters the biophysical properties like increased hydrophobicity of the peptides[73]. Indeed, pyroglutamate-modified Aβ3-x, has been reported to be present in significant fractions in AD[30,74] and exhibits accelerated aggregation kinetics, increased β sheet stability[75] and resistance to degradation in-vivo than a non-modified Aβ peptide[58,76–78]. Together with findings that Aβ3pE-42 and Aβ1-42 peptides form elongated strands that quickly mature to a mesh of long fibrils suggests that Aβ1-42 pyroglutamation seeds Aβx-40 deposition[79]. Moreover, Aβ3pE-x isoforms have been demonstrated to show higher toxicity towards neurons and astrocytes in vitro[79,80], Here, it has been suggested that 3pE isoforms of both Aβ1-42 and Aβ1-40 exert their neurotoxic potential through membrane interactions. Indeed, it has long been discussed that prolonged exposure of neurons to Aβ plaques induces neurotoxic changes through e.g., interaction with membranes[79], receptors[81] or APP[82] itself that could lead to e.g., uncontrolled Ca²⁺ influx[83] and an intracellular response including oxidative stress and radical formation[84]. This can in turn trigger apoptotic mechanisms[79,81] or an autophagy-related cascade of events leading to axonal dystrophy, which is visualized with RTN3[85] as well as PHF-1[86–88]. In addition to Aβ3pE-x, significant changes in levels of Aβ11pE-x peptides were observed that were both present in early, diffuse plaques, both in CUAP and sAD, but increased with plaque maturity in AD compared to non-demented amyloid positives. This is in line with previous pioneering work[35,61–63] though refines those changes to distinct Aβ11pE-x isoforms (Aβ11pE-42 and Aβ11pE-40) across distinct plaque morphotypes in CUAP, sAD and fAD. Similar to Aβ3pE-x isoforms, Aβ11pE-x have been found to enhance aggregation and change fibril morphology[73] as well as show delayed catabolism due to restricted aminopeptidase access[89]. Both suggests that Aβ11pE-x toxicity increases with enhanced Aβ aggregation state similar to Aβ3pE-x isoforms. Of note, the 11E position of Aβ is the primal cleavage site for BACE1, a primary drug target to attenuate beta-amyloid levels. The observation of Aβ11pE-42 and Aβ11-42 in early, diffuse plaques in both AD but also early Downs syndrome and CUAP does not support a plaque-onset related role for BACE1 overactivation beyond a mechanism to combat an initial rise in Aβ levels. In contrast, previous results that showed BACE1 overactivation in AD brain[90]. Along with the current findings that show both higher Aβ11pE-x levels in AD compared to CUAP, as well as continuous Aβ11pE-x accumulation with increased plaque aggregation state, these results do in turn further implicate Aβ11pE-x in AD progression similar to Aβ3pE-x. These conflicting observations highlight the necessity for further investigations on this topic.

Further N-terminal truncations, mainly Aβ4-x, have also been implicated in increased plaque maturation and gain of toxicity. This is thought to occur through similar mechanisms, including reduced solubility, enhanced oligomerization propensity along with decreased clearance and increased brain retention, respectively[91–93]

Together these findings implicate Aβ1-40 deposition along with N-terminal Aβx-42 and Aβx-40 truncation into an amyloid gain of toxicity process. Morphologically, this process involves the maturation of diffuse plaques into cored- and ultimately neuritic plaques. Clinically, this would reflect the transition from a pre-clinical AD, non-demented state to clinical AD.

The multimodal chemical imaging setup further allowed us to identify a distinct dense-cored plaque population that has recently been described as coarse-grained plaque[50]. Coarse-grained plaques stand out as a distinct population of plaques with their unique physical and chemical traits that exhibit dense fibrillar deposition[50]. Typically, coarse-grained plaques are large circular plaques (diameter of ~80 μm) in comparison with typical cored plaques (diameter of ~50 μm). Coarse-grained plaques are devoid of corona while cored plaques are often seen with circular corona around the dense, central deposition, referred to as the core of the plaque.

In our analysis, coarse-grained plaques were observed in both sporadic and familial AD. Notably, the load of coarse-grained plaques was higher in fAD than in sAD, which is in line with previous data, where coarse-grained plaques were more prominent in early-onset AD (EOAD) rather than in late-onset (LOAD) patients[50]. Coarse-grained plaques are observed in both APOE-E4 and -non E4 AD cases but are more prominently seen in homozygous e4 carriers, something which was also observed for sAD in the current study[50].

Histologically, coarse-grained plaques further showed increased levels of neuritic dystrophy, which is relevant as the load of neuritic, dense-core plaques has been shown to correlate with the clinical presentation of dementia[24,94]. Further, in the present study, no neuritic plaque pathology and most prominently no coarse-grained plaque pathology was observed in the non-symptomatic patients with advanced plaque pathology. Both findings suggest that coarse-grained plaques are the key neurotoxic amyloid entities driving disease pathogenesis. The neurotoxic properties of coarse-grained plaques can be related to their specific amyloid makeup, specifically the extensive pyro-glutamation pattern.

The MSI experiments reported here delineated N-terminal isoforms with C-terminal specificity and showed that coarse-grained plaques are predominantly composed of Aβ1-40 and Aβx-40 species, along with smaller proportions of Aβx-42 as compared to, classic cored plaques something that is not discernable with antibodies and IHC. Specifically, the MSI data revealed that coarse-grained plaques showed a high degree of Aβ3pE-40 rather than Aβ3pE-42. Similar to AβxpE-42, pyro-glutamation of Aβx-40 species results in enhanced seeding ability and accelerated aggregation kinetics. Of note, Aβ3pE-40 shows significantly higher increased cytotoxicity in vitro as compared to Aβ3pE-42[79]. This is related to the intrinsic structural properties of Aβ3pE-40 leading to formation of shorter, small, globular aggregates that can easier diffuse through the cell membrane[79]. This effect is further enhanced by the prolonged stability through delayed catabolism[78]. These results highlight the pathogenic relevance of pyroglutamation in AD plaque pathology and provide a further link between plaque pathogenicity and Aβ architectures as further supported by the recent positive clinical trial results for the Aβ3pE-x targeting antibody Donanemab[10].

Of note, our results show that the Aβ pattern associated with coarse-grained plaques closely resembles that of CAA deposits. This finding is interesting, as CAA load has also been associated with clinical dementia both in AD but also familial British and Danish dementia[16,95,96]. Indeed, a vascular component in coarse-grained plaque formation has previously been proposed, where coarse-grained plaques were found to display a characteristic increase in vascular pathology markers, including norrin, laminin and collagen IV, respectively[50]. Herein, we observed higher levels of AβxpE-42 but not Aβ3pE-40 in coarse-grained plaques compared to CAA, indicating the significance of AβxpE-42 in parenchymal plaque formation as opposed to passive Aβ1-40 and Aβx-40 deposition proposed to happen in the vasculature[16,33]. Concurrently, higher levels of Aβ1-40 were noted in CAA compared to coarse-grained-plaques, implying a potential association with vascular permeability or leakage due to the shorter, more globular fibre morphology of Aβ1-40 oligomers[79]. These data, for the first time describe the overlap in Aβ profile between coarse grain plaques and CAA. The differential content in AβxpE-42 provides an explanation for the questions raised why N-terminal Aβ fragments in CAA are able to cross the blood brain barrier while they are not in coarse-grained plaques leading up to parenchymal retention and deposition[50].

Together, the increased degree of N-terminal pyroglutamate truncations and Aβx-40 deposition along with a more pronounced PHF positive neuritic profile compared to cored plaques and the clear association with symptomatic onset and APOE4 status suggest that coarse-grained plaques are a more mature plaque stage implicated with advanced stages of AD.

The here presented AI enhanced Aβ plaque imaging paradigm will open new possibilities for evaluating the in-situ effects of drugs that aim to modify amyloid secretion, reduce its abundance or target distinct isoforms or aggregation intermediates, including e.g., secretase inhibitors or monoclonal antibodies. Moreover, this will also allow to precisely characterize animal models, since those were in the past shown to not accurately mimic biochemical and pathophysiological characteristics of human AD pathology[92,97]. By employing this functional plaque imaging approach in animal model systems will allow to refine plaque type specific biochemical patterns towards functional measures and cellular activation states as well as integrate those observations with new spatial biology techniques.

Further, using the LCO/MSI paradigm will allow to evaluate and further advance amyloid PET tracer development as the method allows to correlatively study plaque polymorphism in autoradiography based ligand binding experiments. Finally, using appropriate pathology cohorts, preferably with available ante-mortem CSF/blood samples, might allow to develop fluid biomarkers that reflect and predict plaque type specific biochemical patterns in the periphery[98].

A limitation of this study is that it is based on a relatively small, cross-sectional post-mortem tissue cohort including solely one brain region per patient, which cannot provide insights into disease progression dynamics. Further, the current study involves a heterogenous group of patients, including fAD individuals with different PS1 mutations with only one case of cotton wool plaque pathology. To address the limitation of static observations, dynamic techniques allowing to follow protein turnover and deposition, like stable isotope labelling kinetics (SILK) are required. While this is naturally difficult to do in human brain tissue, data in transgenic mouse models show promising results for these techniques to elucidate plaque formation dynamics[38,99].

A technical limitation of the MALDI MSI protocol used here is its insensitivity towards phosphorylated peptide species. This is of interest as increased levels of Aβ phospho-Ser8 (Aβ8pS) along with Aβ3pE-x were previously associated with progressing plaque pathology in symptomatic but not preclinical AD[31]. Further, the deep learning was performed on a comparably small sample size. Future work will need to involve benchmarking against state-of-the-art segmentation models to provide greater context and validation for the here presented approach.

In summary, we demonstrate an integrated, AI enhanced, chemical imaging paradigm to interrogate the biochemical signature of morphologically distinct amyloid plaque pathology in AD. Amyloid maturation has previously been implicated in AD pathogenesis proposing that the formation of dense-core plaques is critical in AD pathology. The present data suggests that cored plaques, and more specifically coarse-grained plaques represent a pathological signature indicative of advanced AD.

The results and methods described here are relevant to AD research and drug development, particularly in the light of recent FDA approvals for new biomedicines focusing on the reduction of amyloid plaque load in AD patients, which are believed to target differentially modified and aggregated forms of Aβ. Given that the effect of those drugs has recently only been studied with established biochemical techniques, it is crucial to further understand which structures are targeted and inhibited by these drugs to refine these therapeutic strategies and minimize adverse effects in the future.

## Methods

### Ethics
Ethical approval for the study was obtained from the Local Research Ethics Committee of the National Hospital for Neurology and Neurosurgery, London, UK, as well as the Ethics Review Board at the

University of Gothenburg (Gothenburg, 04/16/2015; DNr 012-15). All studies abide by the principles of the Declaration of Helsinki. The Queen Square Brain Bank (QSBB) operates under ethical approval that permits the collection, storage, and use of donated tissue and associated clinical data for research purposes. This includes the use of de-identified data in research projects.

## Patient samples

Frozen brain tissue samples were obtained from temporal cortex of individuals who had been clinically and pathologically diagnosed with sporadic AD (sAD, $n = 12$) and autosomal dominantly inherited familial AD (fAD, $n = 6$). In addition, $n = 5$ cognitively unimpaired amyloid positive patients were included (Tab. 2). All cases were collected with obtained consent through the brain donation program of the Queen Square Brain Bank for Neurological Disorders (QSBB), UCL Queen Square Institute of Neurology, London, UK. The standard diagnostic criteria were used for the neuropathological diagnosis of AD[2,60,100,101].

## Chemicals and reagents

All chemicals for matrix and solvent preparation were pro-analysis grade and obtained from Sigma-Aldrich/Merck (St. Louis, MO, USA) unless otherwise specified. TissueTek optimal cutting temperature (OCT) compound was purchased from Sakura Finetek (Cat.#: 4583, AJ Alphen aan den Rijn, The Netherlands). Deionized water was obtained by a Milli-Q purification system (Millipore Corporation, Merck, Darmstadt, Germany).

## Tissue Sample Preparation

Frozen tissue sections (12μm) were collected on a cryostate microtome (Leica CM1950, Leica, Bieberach, Germany) and thaw mounted onto indium tin oxide (ITO) conductive glass slides (Bruker Daltonics, Bremen, Germany) and stored at -80 °C. Prior to analysis the sections were thawed and dried under vacuum for 15 min. For protein fixation and lipid removal, the sections were then subjected to a series of sequential washes of 100% EtOH (60 s), 70% EtOH (30 s), Carnoy's fluid (6:3:1 EtOH/CHCl₃/acetic acid) (90 s), 100% EtOH (15 s), H₂O with 0.2% TFA (60 s), and 100% EtOH (15 s).

## LCO Microscopy

For amyloid microscopy, the sections were incubated with two luminescent conjugated thiophene fluorophores (LCO, tetrameric formyl thiophene acetic acid, q-FTAA, 2.4 μM in MilliQ water and heptameric formyl thiophene acetic acid, h-FTAA, 0.77 μM in MilliQ water[102]), in the dark for 25 min. Subsequently, the sections were subjected to a single 10 min 1x PBS wash and stored in the dark at RT until fluorescent microscopy analysis. Multichannel imaging of immuno-stained human brain sections was performed using an automatic widefield microscope (Axio Observer Z1, Zeiss, Germany). Large multi-channel tile scans were captured using EGFP filter sets. All the images were captured using a Plan-Apochromat 20×/0.8 DIC air objective lens.

## MALDI MS imaging

Following microscopy, tissues were subjected to formic acid vapor for 20 min. A mixture of 2,5-dihydroxyacetophenone (2,5-DHAP) and 2, 3, 4, 5, 6-pentafluoroacetophenone (PFAP) was used as matrix compound and applied using HTX TM-Sprayer (HTX Technologies LLC, Carrboro, NC, USA). A matrix solution of 5.7 μl/ml of PFAP and 9.1 mg/ml of DHAP in in 70% ACN, 2% acetic acid / 2% TFA was sprayed onto the tissue sections using the following instrumental parameters: nitrogen flow (10 PSI), spray temperature (75 °C), nozzle height (40 mm), eight passes with offsets and rotations, and spray velocity (1000 mm/min), and isocratic flow of 100 μl/min using 70% ACN as pushing solvent.

MALDI-MSI experiments were acquired on a rapifleX Tissuetyper MALDI-TOF/TOF instrument (Bruker Daltonics) using the FlexImaging and FlexControl (v5.0, Bruker Daltonics) software. Measurements were performed at 10μm spatial resolution, at a laser pulse frequency of 10 kHz with 220 shots collected per pixel. The global laser offset was 93%. Data were acquired in linear positive mode (LP) over a mass range of 1500–6000 Da (mass resolution: m/Δm=1000 (FWHM) at m/z 4515.6).

Pre-acquisition calibration of the system was performed using a combination of peptide calibration standard II and protein calibration standard I (Bruker Daltonics) to ensure calibration over the entire range of potential Aβ species.

Additional data were collected in reflective positive (RP) mode at 10 and 20 μm spatial resolution for comparison of sensitivity (Supplementary Fig. 11). Data were acquired with a laser pulse frequency of 5 kHz, with 220 shots collected per pixel over a mass range of 1500–6000 Da (mass resolution: m/Δm=17000 (FWHM) at m/z 4512.27). Pre-acquisition calibration of the system was performed using a peptide calibration standard I and protein calibration standard I (Bruker Daltonics).

## Immunoprecipitation LC-MS/MS based peptide validation

Tissue homogenization, immunoprecipitation and LC-MS/MS based peptide identification was performed according to a previously reported protocol[103]. Frozen, cortical brain tissue (150 mg) was homogenized in (Tris)-buffered saline (TBS), pH 7.6 containing protease inhibitor (Roche), for 4 min at 30 Hz using a TissueLyzer II (Qiagen). After centrifugation (31,000 × g for 1 h at 4 °C), the supernatant (TBS fraction) was aliquoted and stored at −80 °C. The pellet was resuspended and homogenized in 1 ml of 70% (v/v) formic acid (FA) through sonication. After sonication for 30 s and centrifugation at 31,000 × g for 1 h at 4 °C, the supernatant (FA fraction) was dried in a vacuum centrifuge. Aβ peptides were immunoprecipitated from both fractions using two mouse monoclonal antibodies in combination: 6E10 and 4G8 (cat: 803,003 and 800,711, respectively, BioLegend). Four μg of each antibody was independently conjugated with 50 μl IgG-coated magnetic beads (Dynabeads M−280 Sheep anti-mouse, cat: 11202D, Thermo Fisher Scientific) according to manufacturer's procedures. After conjugation with the beads, the two antibodies were combined and added to each sample. Prior to immunoprecipitation (IP), the dried FA fraction was reconstituted in 200 μL 70% FA (v/v) and shaken for 30 min at room temperature. The reconstituted samples were then centrifuged at 31,000 × g for 1 h at +4 °C and the supernatant was moved to a new tube and neutralized with 4 mL of 0.5 M Tris buffer. TBS and FA reconstituted fractions were incubated over night at +4 °C with antibody-conjugated beads in 0.2% Triton X−100 (w/v) solution. The day after, the samples were washed and Aβ peptides eluted in 100 μl 0.5% FA (v/v). Eluates were dried down in a vacuum centrifuge and stored at −80 °C pending MS analysis. Samples were normalized according to sample volume (i.e., the same volume of brain extract was used for immunoprecipitation).

Amyloid peptides were analyzed with liquid chromatography–tandem MS (MS/MS) analysis using alkaline mobile phase on an UltiMate 3000 binary pump, column oven, and autosampler (Thermo Scientific, Waltham, MA, USA) hyphenated to a Q Exactive quadrupole-orbitrap hybrid mass spectrometer equipped with a heated electrospray ionization source (HESI-II) (Thermo Scientific). The Q Exactive operated in data-dependent mode with resolution settings: 70,000, and target values of $1 \times 10^6$ both for MS and MS/MS acquisitions. Acquisitions were performed with one micro scan per acquisition. Precursor isolation width was 3 m/z units, and ions were fragmented by so-called higher-energy collision-induced dissociation at a normalized collision energy setting of 25[104].

## Immunohistochemistry

For immunohistochemistry and fluorescent amyloid imaging, sections on super frost glasses were fixed in gradient concentration of ice cold 95%, 70% ethanol and 1xPBS at room temperature. Sections were then blocked with (Bovine Serum Albumin BSA, Normal Goat Serum NGS,

Triton in 0.1% PBS-T) for a period of 90 min at room temperature. The sections were then incubated with cocktail of two primary antibodies PHF-1 (Courtesy of Dr. Peter Davies, Feinstein Institute for Medical Research, Manhasset, NY, USA) and RTN3 (Merck Millipore, Catalog # ABN1723) or 6E10 (anti-amyloidβ 1-16) (BioLegend, Catalog # 803003) (diluted in PBS-T with 0.2% NGS (6E10: 1:500, PHF-1: 1:500 and RTN3: 1:1000) for over 18 h at 4 °C. Sections were washed with 0.1% PBST and incubated with secondary antibodies, Alexa Fluor 594 (for PHF-1, Thermo Fisher Scientific, Catalog # A32740) and/or Alexa Fluor 647 (for RTN3, Thermo Fisher Scientific, Catalog # A32728), for a duration of 60 min at room temperature. All the brain sections were then treated with autofluorescence quenching agent TrueBlack™ 1X (Biotium, Fisher Scientific, Catalog # NC1125051) for 30 sec and were later subjected to three 1XPBS washes of 5 min each. To stain amyloid and tau morphologies, antibody-stained sections were incubated with two previously validated LCO fluorophores (tetrameric formyl thiophene acetic acid, q-FTAA, 2.4 μM in MilliQ water) and heptameric formyl thiophene acetic acid (h-FTAA, 0.77 μM in MilliQ water), in the dark for 25 min. Subsequently, the sections were subjected to a single 10 min 1x PBS wash followed by mounting with DAKO fluorescent mounting media and incubated in the dark for 24 h until further imaging.

Multichannel fluorescence imaging of immuno-stained human brain sections was performed using an automatic widefield microscope (Axio Observer Z1, Zeiss, Germany). Large multi-channel tile scans were captured using the EGFP, Alexa Fluor 594 and Alexa Fluor 647 filter sets. All images were captured using the Plan-Apochromat 20×/0.8 DIC air objective lens. The acquisition settings were adjusted to prevent saturation or bleed-through during the acquisition of multichannel images acquired with EGFP/Alexa Fluor 594 / Alexa Fluor647 filter sets.

For microscopy Image Analysis, images were post processed using FIJI ImageJ. Here, files from each channel were split into gray scale images and were subjected to background subtraction. Here EGFP/Green channel with LCO stained plaques are used as a reference image for marking the regions occupied by plaques. Segmentation was performed on the green channel images using the Li threshold method. The Wand tool was used to mark the segmentation as region of interest, which was saved to ROI manager. These ROI annotations were used to acquire intensity measurement from the corresponding gray scale image of anti PHF-1(Alexafluor 594 channel) and anti RTN3 (Alexa Fluor 647 channel).

The obtained data were compiled in MS excel. Fluorescence intensities were calculated as intensity/$\mu m^2$.

## Statistics & reproducibility

**Study design.** Brain tissues from 23 patients were investigated in this study including sporadic AD (sAD, $n = 12$); autosomal dominantly inherited familial AD (fAD, $n = 6$) and cognitively unimpaired amyloid positive patients (CUAP, $n = 5$) (Tab. 2). The number of technical replicate analyses (tissue sections) per patients was $N = 2$. Intra and inter patient variation was determined within the respective groups and across the technical replicates. Further, the specific technical variance of the MALDI MSI technique was estimated by MSI analysis of three sequential sections from one patient. No statistical method was used to predetermine sample size. No data were excluded from the analyses. The experiments were randomized and investigators were blinded to allocation during the experiments and outcome assessment.

## Deep learning model for plaque identification

**Data collection and preprocessing.** A dataset of LCO-based fluorescent amyloid plaque images was collected, ensuring a diverse representation of plaque types. Annotations of the amyloid plaque images were classified into coarse, cored, and diffuse classes based on morphology. The model was trained and tested using a dataset of

annotated amyloid plaque images ($n = 385$), consisting of coarse-grained plaques ($n = 42$), cored plaques ($n = 208$), and diffuse plaques ($n = 135$). The images were preprocessed to standardize size ($120 \times 120$), adjusted by contrast, and converted to a depth of 3 wavelength channels, corresponding to $120 \times 120 \times 1$. This preprocessing helped minimize variations that could affect model performance. The input image resolution and multiple wavelength channel details are also documented in the code, and available upon request.

**Network architecture.** We designed a fully connected neural network (CNN) consisting of multiple convolutional layers followed by max-pooling layers for feature extraction. Batch normalization and dropout layers were employed to enhance model generalization and prevent overfitting. The final classification layer utilized softmax activation to output probabilities for each plaque type.

**Training procedure: three-fold cross-validation.** From the 385 images, 20% ($n = 78$) were set aside as an unseen test dataset. The remaining 307 images were used for training and validation. In each iteration of the cross-validation process, the dataset is split into three mutually exclusive subsets. Two subsets (80% of the data) were utilized for training, while the remaining subset (20%) served as the validation set. Importantly, no data points from the validation set were included in the corresponding training set for that iteration, ensuring that each image was used for validation only once. The predictions of the three trained models were averaged to create a final model. This approach enhanced the robustness of the final model by preventing any overlap between training and validation sets.

**Model training.** The classification model was trained on each combination of training and validation subsets, repeated three times, covering all possible combinations. We employed the Adam optimizer with a learning rate of 0.001 and a learning rate decay schedule (decay factor of 0.96 every 10 epochs) to ensure efficient convergence towards optimal weights. The loss function used was categorical cross-entropy, and we used a mini-batch size of 32 for training. Data augmentation techniques, such as random rotation, flipping, and scaling, were applied to augment the training set, to improve model robustness and to reduce overfitting.

**Evaluation metrics.** The performance of the trained model was evaluated using metrics such as accuracy, precision, recall, and F1-score on the held-out unseen test set ($n = 78$). Confusion matrices were analyzed to assess the model's ability to distinguish between different plaque types.

**MSI data image segmentation analysis.** To identify the localization pattern associated with single plaques, each entire MALDI MSI dataset was interrogated through spatial segmentation using a bisecting k-means based cluster analysis (bkm-CA) algorithm implemented in the SCiLS software (SCiLS Pro v 2023, Bremen, Germany) using the following parameters: Total Ion Count (TIC) normalization and medium smoothing strength. LCO positive amyloid plaques were identified from tile scan images obtained from fluorescent widefield microscopy across the whole cortical brain sections (approximately 1 cm²) for each tissue sample prior to MSI analysis. Segmented MSI cluster images were co-registered with LCO stained images to identify plaque polymorph associated peptide patterns in the MSI data. Pseudo-coloring was done by the software on a pixel to pixel basis and provide an initial visual representation of the chemical differences encoded in different pseudoclusters. Any manual assessment was based on the LCO overlay stratifying the pseudoclusters towards different plaque types identified in the LCO microscopy. Dendrogram cuts were carried out in SciLS to identify whether the segmentation can distinguish those plaques.

**MSI data ROI data analysis.** For bottom-up statistical analysis, deep learning identified plaque morphotypes. LCO images and MALDI data were co-registered in ImageJ and regions of interest (ROI) comprising annotated plaques were exported in FlexImaging (v5.1, Bruker Daltonics). Data from 5-15 plaques per subtype across all patient samples within each group were exported as *.CSV files and imported into R. The raw spectra were processed by means of square root intensity transformation (Sqrt) followed by Savitzky Golay smoothing (Half Window Size = 10). Baseline correction was performed with 100 iterations using the Statistics-sensitive Non-linear Iterative Peak-clipping algorithm (SNIP). Estimation of noise was based on median absolute deviation (MAD). This was followed by total ion count normalization. Peaks were detected based on signal to noise (SNR > 2) and the MAD algorithm. The data were compiled using the R package *MALDIQuant* and deposited in ShinyApp. The AUC values were determined using an in-house created function, based on the trapezoidal rule, which estimates the area by summing the contributions of multiple trapezoids formed by pairs of adjacent data points.

**MSI statistical analysis.** Comparative analysis of peak AUC data was performed for a consistent set of Aβ peptides that were detected in all plaques across all groups that were compared. For multivariate analysis, OPLS-DA analysis and hierarchical cluster analysis (HCA) of averaged peptide intensities (heat maps) was performed using Metaboanalyst 5.0 (https://genap.metaboanalyst.ca/)[105,106]. (HCA parameters: Data: normalized, Standardization: Autoscale features; Clustering method: Ward, Distance: Euclidian; Colour contrast: default.) The data did not follow a normal distribution, as assessed by Shapiro-Wilk test. Therefore, non-parametric statistical tests were used. Following the multivariate analyses, peak AUC data from the different plaque-ROI spectra were analyzed by grouped, univariate statistical comparison for the respective plaque subtypes across patients using Kruskal Wallis ANOVA and Dunn's post hoc multiple comparison test (two-sided) in Prism (v.9, GraphPad, San Diego, CA, USA).

For two-group comparisons, Wilcoxon signed-rank test was used for paired analysis, and Mann-Whitney U test was used for unpaired analysis. To control for multiple comparisons, p-values were adjusted using the Benjamini-Hochberg (BH) procedure. Further, a cumulative intra-patient CV (16%) and inter-patient CV (18%) was calculated across all plaque types and patients. A cumulative technical CV of 10% was calculated across three technical replicates. Multivariate correlations were performed in JMP (v18, JMP Statistical Discovery LLC, Cary, NC, USA) where Pearson Product-Moment Correlations were calculated using a Row-wise method.

### Reporting summary
Further information on research design is available in the Nature Portfolio Reporting Summary linked to this article.

## Data availability
All raw and processed MSI derived ROI MS data are available at: https://maciejdulewiczgu.shinyapps.io/Check_my_SpectRa/. The LC-MS/MS data generated in this study have been deposited in the PRIDE database under accession code PXD060410 Source data are provided with the manuscript. Source data are provided with this paper.

## Code availability
The full code and model weights of the deep learning are publicly accessible at https://github.com/meerask16/plaque_classifier. The script entitled Plaque Classification Model Training, includes a step-by-step implementation of a deep learning pipeline for classifying amyloid plaque images into three categories: coarse-grained, core and diffuse. It includes data preprocessing, model training using a convolutional neural network (CNN) and K-Fold cross-validation.

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

## Acknowledgements

We thank the staff at Centre for Cellular Imaging (CCI), Core Facilities, The Sahlgrenska Academy, University of Gothenburg, for microscopy expertise. The LCO probes were developed and donated as a kind gift from Prof. Peter Nilsson and Prof. Per Hammarström, Linköping University, Sweden. The Queen Square Brain Bank is supported by the Reta Lila Weston Institute of Neurological Studies, UCL Queen Square Institute of Neurology.

J.H. is supported by the NIH-NIA (R01 AG078796, R21AG078538, 1R21AG080705), the Swedish Research Council VR (#2019-02397, #2023-02796), the Swedish Alzheimer Foundation (#AF-968238, #AF-939767, #AF994082), the Swedish Brain Foundation Hjärnfonden (FO2022-0311), Magnus Bergvalls Stiftelse and Åhlén-Stiftelsen (#233011, # 243090). Stiftelsen Gamla Tjänarinnor (S.K., J.H., W.M., H.Z., K.B.) and Gun och Bertil Stohnes Stiftelse (J.H., S.K., W.M.) are acknowledged for financial support. H.Z. is a Wallenberg Scholar and a Distinguished Professor at the Swedish Research Council supported by grants from the Swedish Research Council (#2023-00356; #2022-01018 and #2019-02397), the European Union's Horizon Europe research and innovation program under grant agreement No 101053962, Swedish State Support for Clinical Research (#ALFGBG-71320), the Alzheimer Drug Discovery Foundation (ADDF), USA (#201809-2016862), the AD Strategic Fund and the Alzheimer's Association (#ADSF-21-831376-C, #ADSF-21-831381-C, #ADSF-21-831377-C, and #ADSF-24-1284328-C), the Bluefield Project, Cure Alzheimer's Fund, the Olav Thon Foundation, the Erling-Persson Family Foundation, Familjen Rönströms Stiftelse, Hjärnfonden, Sweden (#FO2022-0270), the European Union's Horizon 2020 research and innovation program under the Marie Skłodowska-Curie grant agreement No 860197 (MIRIADE), the European Union Joint Program – Neurodegenerative Disease Research (JPND2021-00694), the National Institute for Health and Care Research University College London Hospitals Biomedical Research Centre, and the UK Dementia Research Institute at UCL (UKDRI-1003). K.B. is supported by the Swedish Research Council (#2017-00915), the Alzheimer Drug Discovery Foundation (ADDF), USA (#RDAPB-201809-2016615), the Swedish Alzheimer Foundation (#AF-930351, #AF-939721 and #AF-968270), Hjärnfonden, Sweden (#FO2017-0243 and #ALZ2022-0006), the Swedish state under the agreement between the Swedish government and the County Councils, the ALF-agreement (#ALFGBG-715986 and #ALFGBG-965240), the European Union Joint Program for Neurodegenerative Disorders (JPND2019-466-236), the National Institute of Health (NIH), USA, (grant #1R01AG068398-01), and the Alzheimer's Association 2021 Zenith Award (ZEN-21-848495). T.L. is supported by an Alzheimer's Research UK senior fellowship. Queen Square Brain Bank is supported by the Reta Lila Weston Institute for Neurological Studies.

## Author contributions

S.K. Performed LCO and IHC microscopy experiments, LCO and IHC image analysis and designed parts of the Figures. J.G. Performed all MALDI analyses, all multivariate MSI data analysis, the IPMS-LCMSMS experiments and data analyses, designed parts of the Figures. M.D. Processed and analyzed all MSI ROI spectral data, performed all statistical analyses and critically reviewed the manuscript; M.S. developed the deep learning code and trained the model. A.S. analyzed the data and critically reviewed the manuscript, J.W. developed the plaque radii image analysis tool and critically reviewed the manuscript, K.B. provided financial support, critically analyzed and discussed the data, reviewed the manuscript, H.Z. provided financial support, critically analyzed and discussed the data, reviewed the manuscript. K.B. provided financial support, critically analyzed and discussed the data, reviewed the manuscript W.M. developed parts of the methodology and critically analyzed the data, N.R. evaluated cases from the QSBB and critically reviewed the manuscript. T.L. selected the cases from QSBB archives and critically reviewed the manuscript. J.N.S provided significant input on all aspects of the study, discussed and analyzed the data, critically reviewed the manuscript, M.Sch. developed the DL methodology together with M.S, critically analyzed and discussed the results, reviewed the manuscript, J.H. conceived and designed the study, analyzed the data, designed the Figures, provided financial support and wrote the manuscript.

## Funding

## Competing interests

H.Z. has served at scientific advisory boards and/or as a consultant for Abbvie, Acumen, Alector, Alzinova, ALZPath, Amylyx, Annexon, Apellis, Artery Therapeutics, AZTherapies, Cognito Therapeutics, CogRx, Denali, Eisai, Merry Life, Nervgen, Novo Nordisk, Optoceutics, Passage Bio, Pinteon Therapeutics, Prothena, Red Abbey Labs, reMYND, Roche, Samumed, Siemens Healthineers, Triplet Therapeutics, and Wave, has given lectures in symposia sponsored by Alzecure, Biogen, Cellectricon, Fujirebio, Lilly, Novo Nordisk, and Roche, and is a co-founder of Brain Biomarker Solutions in Gothenburg AB (BBS), which is a part of the GU Ventures Incubator Program (outside submitted work). KB has served as a consultant, at advisory boards, or at data monitoring committees for Abcam, Axon, BioArctic, Biogen, JOMDD/Shimadzu. Julius Clinical, Lilly, MagQu, Novartis, Ono Pharma, Pharmatrophix, Prothena, Roche Diagnostics, and Siemens Healthineers, and is a co-founder of Brain Biomarker Solutions in Gothenburg AB (BBS), which is a part of the GU Ventures Incubator Program, outside the work presented in this paper. The remaining authors declare no competing interests.

## Additional information

[1]Department of Psychiatry and Neurochemistry, Sahlgrenska Academy, University of Gothenburg, Gothenburg, Sweden. [2]Wallenberg Centre for Molecular
and Translational Medicine, University of Gothenburg, Gothenburg, Sweden. [3]Department of Neuroscience, Physiology and Pharmacology, University
College London, London, UK. [4]Clinical Neurochemistry Laboratory, Sahlgrenska University Hospital Mölndal, Mölndal, Sweden. [5]Paris Brain Institute, ICM,
Pitié-Salpêtrière Hospital, Sorbonne University, Paris, France. [6]Neurodegenerative Disorder Research Center, Division of Life Sciences and Medicine, and
Department of Neurology, Institute on Aging and Brain Disorders, University of Science and Technology of China and First Affiliated Hospital of USTC,
Hefei, PR China. [7]Department of Neurodegenerative Disease, Queen Square Institute of Neurology, University College London, London, UK. [8]UK Dementia
Research Institute, University College London, London, UK. [9]Hong Kong Centre for Neurodegenerative Diseases, Hong Kong, China. [10]Wisconsin Alzheimer's
Disease Research Center, University of Wisconsin School of Medicine and Public Health, University of Wisconsin-Madison, Madison, WI, USA. [11]Department of
Public Health and Caring Sciences, Uppsala University, Uppsala, Sweden. [12]Dementia Research Centre, Queen Square Institute of Neurology, University
College London, London, UK. [13]Queen Square Brain Bank for Neurological Disorders, Department of Clinical and Movement Neurosciences, Queen Square
Institute of Neurology, University College London, London, UK. [14]Ken and Ruth Davee Department of Neurology, Northwestern University Feinberg School of
Medicine, Chicago, IL 60611, USA. [15]Department of Neuropsychiatry, Sahlgrenska University Hospital, Gothenburg, Sweden. [16]These authors contributed
equally: Srinivas Koutarapu, Junyue Ge, Maciej Dulewicz. ✉e-mail: jh@gu.se

