## [Transparent Peer Review file · Nature Communications]

Chemical imaging delineates A β plaque polymorphism across the Alzheimers disease spectrum

Corresponding Author: Dr Jorg Hanrieder

Version 0:

Reviewer comments:

Reviewer #1

(Remarks to the Author)

Koutarapu et al. used MALDI mass spectrometry (MS) imaging to determine the composition of different types of amyloid plaques in the Alzheimer's disease (AD) brain. Only symptomatic AD cases were included in this study, 9 sporadic AD cases and 6 familial AD cases with PSEN1 mutations. Asymptomatic AD cases were not included in this study! In addition to MALDI MS imaging, immunohistochemistry and staining with LCOs was performed. Main findings were the distinction between diffuse plaques, cored, and coarse-grained plaques based on the composition of different Abeta forms including 3pE and 11pE forms as seen by MALDI MS imaging. This information was used to train a model algorithm for MALDI MS imaging-based plaque detection. Statistical analysis was based on the number of plaque images taken without further respect on case-related variance. The main differences among the plaque types were: diffuse plaques contained less Abeta40 and pE3Abeta than cored and coarse-grained plaques. Cored plaques differed from coarse-grained plaques in the amount of Abeta40 and 3pE species with coarse-grained plaques having a similar composition as CAA-related Abeta deposits. The authors claim that this is the first study linking biochemical characteristics of amyloid plaque polymorphism.

Using MALDI MS imaging for analysis of amyloid plaques is novel. However, the results that were obtained just confirm earlier immunohistochemical studies on Abeta species in amyloid plaques (PMID: 8952519, 7857653, 9173910, 32926214, 24519982). Therefore, the main novelty in this study as presented here is the method.

Technically, the experiments are properly carried out. All experiments and results are well described. The discussion can be improved (see concerns).

Concerns:

- The number of cases enrolled in this study is with 9 AD and 6 familial AD cases low. Statistical analysis does not seem to take inter-individual variation into account when pooling multiple plaques from just a few cases. A statistical method need to be used that allows adjustment for inter-individual variations among the different cases. It could be, for example, that the majority of diffuse plaques was seen in few cases and were compared to coarse grained plaques from other cases. If so, the differences in the plaque content would reflect the genetics of the individuals rather than differences in the composition of different plaque types.
- Coarse grained plaques were initially reported to be restricted to early onset AD cases. How is this plaque type represented among all cases? Is the definition of coarse-grained plaques 100 % equal to what was reported by Boon et al. (PMID: 32926214) or does MALDI MS imaging extends the definition?
- The authors claim novelty in the delineation of diffuse and fibrillar plaques and CAA. As far as I can see they just confirm known facts based on an extended dataset of Abeta species. The findings about pE3 and pE11 Abeta fit with published findings. I don't see novel real insights into plaque biology yet.
- The study has the limitation that only symptomatic AD cases were included that represented advanced stages of the disease. Early stages of plaque development are not covered but would be important, especially when speculating about plaque maturation. Given the cases used (all Thal 5), even the diffuse plaques have signatures of advanced AD. Early-stage AD signatures can only be obtained in early asymptomatic AD stages (Thal 1 or 2).
- The term fibrillar plaques appears to be misleading since also diffuse plaques consist of small Abeta fibrils as indicated by silver stainings and electron microscopy (PMID: 1759558, 1950474, 10029103). Probably, it may be better to stick with the terms "cored" and "coarse-grained" plaques.

Reviewer #2

(Remarks to the Author)

Koutarapu et al., performed an interesting study investigating amyloid-beta isoforms in distinct amyloid-beta deposits, including diffuse plaques, cored plaques, coarse-grained plaques, neuritic plaques, and cerebral amyloid angiopathy in both sporadic Alzheimer's disease and familial Alzheimer's disease using MALDI-TOF on frozen tissue. Distinct deposit types were imaged using LCO fluorophores. Plaque recognition was automated using a deep-learning model. The authors find that distinct deposits consist of specific predominant amyloid-beta isoforms. They find similar compositions for abeta peptides in plaques sporadic and familial AD, but differences with regards to content of dystrophic neurites.

I think this is an important study for the field, especially in the era of anti-amyloid therapies. I do have comments that would improve the paper:

General:

- The findings are mostly a confirmation of what is previously shown with IHC, IF, and ELISA by Boon et al., but with a different technique. It would be appropriate for the authors to acknowledge that by rephrasing some wording throughout the manuscript. The new comparison is sporadic versus familial AD.

Methods:

- The case information (Table 1) can be improved:

What is the APOE genotype?

Are the sporadic AD cases screened for autosomal dominant AD mutations (PS1, PS2, APP)?

Typo: ABC score for case sAD9: C score cannot be 4

- The authors developed a deep learning model, but who trained the model? And what was the interrater variability?

The manuscript was somewhat difficult to read.

- The abbreviations for plaques make the manuscript difficult to read for people less familiar with plaques. Please do not abbreviate the plaques throughout the paper, except for in the figures. The appropriate abbreviations in the figures would be the first letter of each word: cored plaque (CP), neuritic plaque (NP), coarse-grained plaque (CGP), cotton wool plaque (CWP)

- Since the coarse-grained plaque versus cored plaque is a major question, please explain these two plaque types in paragraph 2 of the introduction with appropriate references.

- Coarse-grained plaque is misspelled.

- Some information comes later than expected, for example it is unclear which secondary (alexafuor594 or 647) is used for which primary (PHF-1 or RTN3) in the first description of immunofluorescence. This only becomes clear in the text about image analysis.

- Headings can be improved: the text about image analysis needs a different caption than what it currently is under (Immunohistochemistry).

- Not all figures that are referred to, can be found in the supplementary data: for example, I could not find Fig. S8, referred to in the section "Evaluation Metrics:".

- Figures are not always clear, for example

o figure 3(a): What does a VIP of 0.5 mean, is this the ratio of CP/DP?

o figure 3(b): what do the cluster lines mean?

Other

- The coarse-grained plaque is not the same as the primitive plaques or burned-out plaques. These are a different category, please correct this.

- The authors write about plaque maturation and fibrilization, but postmortem tissue is cross sectional, therefore the authors need to be aware this maturation is an assumption.

Reviewer #3

(Remarks to the Author)

Review for "Chemical imaging delineates A β plaque polymorphism across Alzheimer's disease spectrum"

This is a well written paper that tries to shed light on the association of polymorphic A β plaque pathology with AD pathogenesis, clinical symptoms and disease progression. The authors are using advanced imaging tools such as functional amyloid microscopy and matrix assisted laser desorption ionization (MALDI) with TOF/TOF mass spectrometry. Because of the complexity of the data a considerable amount of statistical treatment of the data is done to try to draw out relationships between the observations from the data regarding the various forms of the A β plaques and their pathological relationships including using deep learning algorithms to interpret the mass spectrometry data.

The authors need to define all abbreviations used in the paper including common abbreviations used in statistics. As an example, I couldn't find a definition for VIP score in the paper. Does it refer to variable importance plots?

The authors state that they are using a Bruker rapifleX Tissue type MALDI TOF/TOF. I would have expected a better resolution than 4,500. Also, why were the samples run in linear mode when the TOF/TOF (reflectron) mode will give much better resolution and much improve mass spec data. Did the authors consider the possibility of laser induced fragmentation for some of the A β pieces observed?

The authors need to clearly note how many of the patient samples were analyzed using mass spectrometry. They report having 15 patient samples, 9 sporadic and 6 familial. How many tissue samples from each patient were analyzed using mass spectrometry? What thickness of tissue sample was used? In line 318-19 they state that serial sections were analyzed with both MALDI MSI and fluorescence microscopy. Are the examples in part b and part c Figure 1 examples of specific tissue sections or aggregates of the serial sections, how many serial sections were there for each patient? How reproducible were the mass specs from the serial sections?

The authors need to clearly state what sample(s) were used for the MALDI MSI images presented in Figure 3 part c. If this was from a single tissue specimen, then the authors need to add a picture of the tissue so that readers can be oriented to the tissue morphology. Also, the authors need to state the mass window used to generate the mass heat maps in part c and the mass of each of the polyforms should be included with the appropriate number of significant figures. If the mass heat maps are all from the same tissue specimen, then the authors should include the full mass spectrum. This also applies to the single ion images presented in Figure 5. The authors need to clearly state the number of samples used.

The only mass spectrum shown in the paper is in figure 3 part h and it is averaged. The authors need to clearly state how the average mass spectrum being presented was generated. Were all the specimens used for the average mass spectrum?

The images shown in general are not very clear in how they relate to tissue morphology which is the goal in mass spec imaging (MSI). One of the primary advantages of using MSI is to be able to relate the compounds being observed to the cellular structure of the tissue. The images show no discernible tissue structure.

SI Table S1: Masses of the detected A β isoforms. The authors need to include at least representative mass spectra which show these isoforms. Were they observed in all MSI experiments. For each of the masses the exact mass for each one is calculated to 5 decimal places. Linear mode will not give the necessary resolution to justify this. What is being calculated for the [M+H]⁺ Avg? The values given for AVG look more like nominal masses. What is the purpose of presenting this data? The authors need to update the table using the proper number of decimal places for the masses and report the accurate mass data from the MSI experiments.

The authors do not present a compelling argument for their conclusions using the mass spectrometry data. The mass spectrometry data does not give any insight into whether the A β polymorphism locations are coincidence or whether there is an actual correlation. More careful mass spectrometry work needs to be done rather than applying a battery of statistical tools. I also believe that this holds true for the microscopy data for similar reasons as stated above for the mass spectrometry data. It is unclear what samples are being used.

MSI is used ideally with no specific physiological outcome in mind but being untargeted allowing the biochemistry to shine through. I would expect many other small proteins and peptides to be observed in this mass region (1500 to 6000 Da).

Reviewer #4

(Remarks to the Author)

This work displays effective use of machine learning and deep learning to discover new chemical insights into AD, with extendability to other spectroscopy/MSI tasks and samples. The discussion clearly describes the tangible findings, and statistics are reported to support claims. However, revisions should be made surrounding clarity/transparency of the methods, and importantly the disclosure of code and workflows (i.e. a GitHub), as currently this work is non-reproducible. Additionally, ML terminology can be standardized when appropriate.

Line 121: Briefly explaining why MSI provides an advantage over other techniques would be beneficial, or potentially mentioning how MALDI-MS can quantize various analytes simultaneously with high throughput, etc. rather than stating its one of many modalities, superior to staining, would better provide readers with the context of why the particular method was employed. It may also be beneficial to describe MSI as being an orthogonal approach to staining/spectroscopy since immuno-staining was done.

Could also make sense to have some text and citations regarding DL for image classification and/or MSI in the context of histology etc.

Line 190: The actual architecture should be reported as code for training and with model weights for final inference. Important details such as image depth (i.e. multiple wavelength channels) that are not made explicit in text would be covered by accompanying code (as the figures of the architecture are not comprehensive, nor should they). Details such as the loss function, mini-batch size, learning rate (and decay) should be reported, as well as loss vs epoch plots in the supporting.

Line 196: Its not clear why cross-training is being used as terminology instead of cross-validation.

Line 199 Is it the parameters or predictions that are averaged? There should be accompanying code for the deep learning. Would the authors be employing an ensemble model as their 'final model'?

Line 203: Do the authors mean each pair of training and validation splits? Each combination would be highly irregular, as there would certainly be identical data in both the training and validation for many combinations.

Line 213: The SI only goes to page 10 and ends with Figure S7. Table of contents also states SI Figures S1-7

Line 234: Authors should make clear to readers how HCA was employed, potentially in the SI. Please change 'k mean' to 'k-means'. Code regarding HCA should be available. Is pseudo coloring being done on a pixel by pixel basis by pseudo cluster assignment? Supporting figures only show AB isoform intensities as variables for the HCA, and it would not be clear to many readers how this can lead to segmentation. Any manual assessment of initial clustering/ML required for setting the final parameters for tasks such as dendrogram cuts should be described if employed.

Line 339: Would be interesting to have label probabilities for these misclassified samples to gain insight into the model. Did these misclassifications make visual sense given similarity to other classes? In any case, AUC and ROC curves should be reported.

Line 348: These are indeed great results. Was the model benchmarked against other (plaque) segmentation models? This is a very low amount of data for deep learning, was transfer learning not employed?

Figure 2e seems unnecessary given its description in text.

Further background

Version 1:

Reviewer comments:

Reviewer #1

(Remarks to the Author)

This the revised version of a previously submitted manuscript. The authors sufficiently addressed my points.

(Remarks on code availability)

Reviewer #3

(Remarks to the Author)

The authors have addressed my concerns regarding the original manuscript and have included the appropriate edits. The submission can be published as is.

(Remarks on code availability)

I did not review the code.

Reviewer #4

(Remarks to the Author)

The authors have done a wonderful job addressing many of the requested revisions and questions.

They have shown excellent work and given the availability of their trained model I leave acceptance of the article to the editors.

(Remarks on code availability)

The code is easy to follow, and well organized. Training scripts should be provided, however.

Letter of Response

Please find attached with this resubmission our revised manuscript “Chemical imaging delineates A β plaque polymorphism across Alzheimer's disease spectrum” by Koutarapu et al. We are grateful for the overall positive, thorough and constructive feedback provided by the expert reviewers and editors, those helped significantly to both clarify and improve the manuscript substantially. Please find below a detailed point-by-point response to the comments raised by the reviewers.

The cross-references (page and line numbers) to the corresponding changes made refer to the manuscript with tracked changes (“all markup”) that is provided as supplemental with this resubmission.

With best regards
Jörg Hanrieder

Reviewer #1 (Remarks to the Author):

1. Koutarapu et al. used MALDI mass spectrometry (MS) imaging to determine the composition of different types of amyloid plaques in the Alzheimer's disease (AD) brain. Only symptomatic AD cases were included in this study, 9 sporadic AD cases and 6 familial AD cases with PSEN1 mutations. Asymptomatic AD cases were not included in this study! In addition to MALDI MS imaging, immunohistochemistry and staining with LCOs was performed. Main findings were the distinction between diffuse plaques, cored, and coarse-grained plaques based on the composition of different Abeta forms including 3pE and 11pE forms as seen by MALDI MS imaging. This information was used to train a model algorithm for MALDI MS imaging-based plaque detection. Statistical analysis was based on the number of plaque images taken without further respect on case-related variance. The main differences among the plaque types were: diffuse plaques contained less Abeta40 and pE3Abeta than cored and coarse-grained plaques. Cored plaques differed from coarse-grained plaques in the amount of Abeta40 and 3pE species with coarse-grained plaques having a similar composition as CAA-related Abeta deposits. The authors claim that this is the first study linking biochemical characteristics of amyloid plaque polymorphism.

Using MALDI MS imaging for analysis of amyloid plaques is novel. However, the results that were obtained just confirm earlier immunohistochemical studies on A β species in amyloid plaques (PMID: 8952519, 7857653, 9173910, 32926214, 24519982). Therefore, the main novelty in this study as presented here is the method.

Response: We appreciate the thorough and constructive comments provided. We concur that the methodological aspects of our work are a significant advancement in itself that however open up for novel biological insight.

1A. Regarding conceptual and technical novelty, the MALDI MSI technique comes with the significant advantage of superior chemical specificity, comprehensiveness and integrability with chemical staining. Importantly, this allows to capture all present A β peptide species in the same analysis, on the same tissue and at the same spatial resolution.

This way, it is possible to associate correlation of peptide abundances at the single pixel level, within and across different plaque entities and finally contextualize those with discrete and unbiased structural properties as warranted by the integrated functional microscopy that was enhanced by deep learning (which is a further technical novelty).

To further illustrate those unique capabilities of our chemical imaging paradigm, we include an updated Figure 1 along with a technical workflow Figure S1a.

Further, the molecular specificity of mass spectrometry exceeds the specificity of antibodies which in turn provides new biochemical insight of plaque pathology.

This is best illustrated in two ways, particularly with respect to pyroglutamated isoforms given the emerging relevance of donanemab.

- First, Anti-pE antibodies do not provide full information on the C-terminal identity of the pyro-Glu A β species detected as IHC is not able to distinguish an entire full length peptide and is limited to one (chromogenic detection) or few channels (fluorescent microscopy). Due to the indirect detection, immunohistochemical detected sequences (eg. 3pE-x, x-42) might spatially correlate but this does not provide ultimate proof of correct full sequence identity and localization, sth that is addressed by mass spectrometry imaging. Moreover, due to the sequential nature,

IHC does not provide relative comparison of single species to other full length A β isoforms to estimate global levels of A β specific truncation (eg levels of 3pE-42/1-42). Again, this is sth achievable by MS imaging.

Second, while we acknowledge that some 3pE-x and 11pE-x targeting antibodies show promising IHC/WB data and provided valuable insight on plaque pathology, those antibodies cannot distinguish 3pE-40 and 3pE-42 (PMID: 28623233) or 11pE-40 from 11pE-42 (PMID: 22001577). Indeed, A general limitation for most of the studies describing new anti pE-A β antibodies is that cross reactivity of the pE-antibodies towards non-pE (eg 3-x/4-x vs 3pE-x) or other pE species (3pE-x vs 11pE-x) was not or only in part investigated (PMID: 20864186), sth that has been noted though in the context of British and Danish dementia related amyloid peptide species (Saul et al 2013 PMID: 23261769).

We mention these considerations in the introduction (p5lin4-18) and discussion (p21line 27-p22line18, p26line 4-26)

1B. Regarding biological, novel findings, we agree that many of the detected peptide patterns have been described before in the context of heterogenous plaque pathology and implicated into sequential plaque deposition with disease progression as rightfully mentioned by in your comment above. The seminal work by was now mentioned in more detail in the discussion and introduction.

Regarding the present results, we both confirm these previous findings and even expand this knowledge:

- Here, a first biological novelty lies within that we demonstrate for the first time differences in 3pE and 11pE peptides both x-40 and x-42 across different plaque types such as diffuse, cored, cotton-wool-, coarse grain plaques and CAA that are not discernable using conventional microscopy. (p23line 1-8, p26line4-p27line9)
- Second, we extend these differential plaque analyses across sAD, fAD and CUAP. Here, we show that eg 3pE-42 and 11pE-42 peptides are present in CUAP plaques as well as show that cored plaques in CUAP and sAD are relatively similar, and merely differ with respect to relative abundance of key peptides (1-40, 3pE-42, 1-42) and neuritic content. Of note, we show that CUAP cored plaques demonstrate a similar relation of elevated 3pE-42 and 1-40 deposition as observed in sAD though at lower levels suggesting that cored plaque formation is similar across those pathologies and its aggravation (ie excessive 3pE-42 generation and 1-40 deposition) is associated with the development of AD. (see p23line 7-11)
- Third, this is the first report describing and comparing coarse grained plaques in both sAD and fAD. We show that those are in fact similar in biochemical nature

and their difference in abundance was similar to what was reported by Boon et al regarding plaque load in EOAD and LOAD. Moreover, the absence of CGP formation in CUAP highlights their association with cognitive decline/AD onset. (see p25line19-p26line18)

- Fourth, our work extends Boones work by elucidating characteristic 3pE-42, 11pE-42 and 3pE-40 peptides in CGP as well as by further comparing A β isoforms of CGP with CAA. Here, for instance, A β 3pE-42 and 11pE-42 in CGP and A β 3pE-40 in both CGP and CAA, are in fact, an as-yet unreported (novel) finding in AD. Moreover, we show higher degrees of pyroglutamation in CGP than in CAA along with x-42 peptide deposition. In fact, Boon et al. raised a question in their discussion: "Why the A β x-40 in case of CAA is capable of crossing the blood–brain barrier and in case of the coarse-grained plaque is not, remains elusive." Our results possibly answer that question, given the increased amounts of pyroglutamation and x-42 species in CGP compared to CAA as those species are both more hydrophobic and aggregation prone. (see p26line4-p27line9)

R: We updated these parts in the discussion (see the references above to the corresponding parts in the revised manuscript).

2. Technically, the experiments are properly carried out. All experiments and results are well described. The discussion can be improved (see concerns).
Concerns:

- The number of cases enrolled in this study is with 9 AD and 6 familial AD cases low.

R: We concur that this is a major limitation and have expanded the sAD cohorts with three more patients. In addition, we now provide data for 5 cognitively unaffected amyloid positive individuals (Thal 4).

Please see Figures 3, SI Fig S2+3, Results (p6line18-20) and Discussion, p22line22-27

3. Statistical analysis does not seem to take inter-individual variation into account when pooling multiple plaques from just a few cases.

A statistical method need to be used that allows adjustment for inter-individual variations among the different cases. It could be, for example, that the majority of diffuse plaques was seen in few cases and were compared to coarse grained plaques from other cases. If so, the differences in the plaque content would reflect the genetics of the individuals rather than differences in the composition of different plaque types.

R: We appreciate this concern and have been investigating inter and intra-plaque related variations across all plaque types and amyloid peptides. The cumulative intra-patient variation was <16% and the inter-patient variation was <18%.

We specify this in the methods and results. See P34line24-27

4. Coarse grained plaques were initially reported to be restricted to early onset AD cases. How is this plaque type represented among all cases? Is the definition of coarse-grained plaques 100 % equal to what was reported by Boon et al. (PMID: 32926214) or does MALDI MS imaging extends the definition?

R: This is a valuable point. Please see our detailed comment regarding the biological novelty of extending the biochemical CGP definition above (point 1B, 4th paragraph)

Briefly, our experiments are in line with defining coarse grained plaques with respect to general amyloid patterns (A β x-40, A β x-42, A β N3pE-x), morphology and neurotoxicity (dystrophic neurites)/neuritic components (PHF). Our approach extends this previous definition with regard to structural information (LCO), dystrophy in a plaque proximity related manner and comprehensive detection of full length A β isoforms on a single plaque level.

Of note, this identified specific 3pE-40 vs 3pE-42 differences that e.g. could a) not be picked up with IHC and b) not mapped in the context of other peptides and structural information.

Please see the updated Discussion (see p26line4-p27line9)

5. The authors claim novelty in the delineation of diffuse and fibrillar plaques and CAA. As far as I can see they just confirm known facts based on an extended dataset of Abeta species. The findings about pE3 and pE11 Abeta fit with published findings. I don't see novel real insights into plaque biology yet.

R: We do concur that this part requires clarification. Please see our comment regarding plaque biology above (point 1B). We made a substantial effort to properly acknowledge these well-known previous findings and aimed to demonstrate where our data either confirm or expand current knowledge.

Please see our updated Introduction (p5line3-17, p6line6-11) and Discussion (p23line6-13)

6. The study has the limitation that only symptomatic AD cases were included that represented advanced stages of the disease. Early stages of plaque development are not covered but would be important, especially when speculating about plaque maturation. Given the cases used (all Thal 5), even the diffuse plaques have signatures of advanced AD. Early-stage AD signatures can only be obtained in early asymptomatic AD stages (Thal 1 or 2).

R: We agree with this very important comment. We managed to include brain tissue from 5 individuals who were cognitively unimpaired despite significant amyloid burden (Thal 4). Our rationale for using Thal 4-staged CUAP was that this is a more appropriate comparison towards the symptomatic AD patients as differences in plaque chemistry can then be linked to clinical manifestation in this cross-sectional study, where no longitudinal data were available. We rationalize that if one were to investigate Thal 1 or 2 cases, one could not know where these cases would end up over time, i.e. whether they would end up being symptomatic AD or CUAP cases.

Regarding the results, we present another biological relevant finding where we observe a) less amounts of cored plaques and no CGP in CUAP as compared to sAD (Fig.1c).

b) CP in CUAP showed higher levels of x-40 and 3pE-x peptides as compared for DP, similar though less pronounced to what is observed for cored plaques in symptomatic AD.

Our interpretation is that this ties together 3-42 pyroglutamation and 1-40 deposition as main factors in sequential deposition of cored plaques.

We extended the manuscript accordingly and state this limitation in the discussion.

Please see Figures 3, SI Fig S2+3, Results (p6line18-20) and Discussion (p22line20-27, p24line 10-27)

7. The term fibrillar plaques appears to be misleading since also diffuse plaques consist of small Abeta fibrils as indicated by silver stainings and electron microscopy (PMID: 1759558, 1950474, 10029103). Probably, it may be better to stick with the terms “cored” and “coarse-grained” plaques.

R: We agree that there is no consensus on defining plaque types based on fibril content across the literature as rightfully highlighted by the reviewer since even diffuse deposits show small fibrillar content while others do not observe any fibrils in some DP (MID 1950474).

We acknowledge this in the text when defining diffuse plaques in the introduction and a newly generated Tab.1. We further define the nomenclature towards diffuse vs dense-cored (cored and coarse-grained) and abolished the ‘fibrillized’ terminology as suggested. We have been revising the manuscript accordingly.

Reviewer #2 (Remarks to the Author):

Koutarapu et al., performed an interesting study investigating amyloid-beta isoforms in distinct amyloid-beta deposits, including diffuse plaques, cored plaques, coarse-grained plaques, neuritic plaques, and cerebral amyloid angiopathy in both sporadic Alzheimer's disease and familial Alzheimer's disease using MALDI-TOF on frozen tissue. Distinct deposit types were imaged using LCO fluorophores. Plaque recognition was automated using a deep-learning model. The authors find that distinct deposits consist of specific predominant amyloid-beta isoforms. They find similar compositions for abeta peptides in plaques sporadic and familial AD, but differences with regards to content of dystrophic neurites.

I think this is an important study for the field, especially in the era of anti-amyloid therapies. I do have comments that would improve the paper:

General:

1. The findings are mostly a confirmation of what is previously shown with IHC, IF, and ELISA by Boon et al., but with a different technique. It would be appropriate for the authors to acknowledge that by rephrasing some wording throughout the manuscript. The new comparison is sporadic versus familial AD.

R: We concur with this very relevant comment and updated the introduction and discussion accordingly. We highlight previous achievements and contrast our results to those, while showing where the data are confirmative and where they extend current knowledge.

2. Methods: The case information (Table 1) can be improved: What is the **APOE genotype**?

R: We updated Table 1 (now Table 2). We further include plots for plaque load across all groups/AD subtypes and stratified in fig 1c. We further evaluated plaque loads for group differences dichotomized after etiology and APOE type but did unfortunately not see any statistical difference or correlation. Please see Fig. 1c

3. Are the sporadic AD cases screened for autosomal dominant AD mutations (PS1, PS2, APP)?

R: The majority of sporadic AD cases donated to QSBB come through the Dementia Research Centre, if there was a family history documented then the case would have been screened for the dominant AD mutations. None of the cases used in the study had a family history of dementia. (see Tab 2)

4. Typo: ABC score for case sAD9: C score cannot be 4

We have revised this mistake in the manuscript.

5. The authors developed a deep learning model, but who trained the model? And what was the interrater variability?

The model was trained by one of the authors (MeS). Plaque annotation was rated by two independent neuropathologists that are also co-authors of the paper. As both agreed on classifications, we did not formally investigate inter-rater variability (please see also SI Fig 10).

6. The manuscript was somewhat difficult to read. The abbreviations for plaques make the manuscript difficult to read for people less familiar with plaques. Please do not abbreviate the plaques throughout the paper, except for in the figures. The appropriate abbreviations in the figures would be the first letter of each word: cored plaque (CP), neuritic plaque (NP), coarse-grained plaque (CGP), cotton wool plaque (CWP)

R: Thank you for pointing this out, we have amended the writing accordingly.

7. Since the coarse-grained plaque versus cored plaque is a major question, please explain these two plaque types in paragraph 2 of the introduction with appropriate references.

R: This was revised with appropriate references. Please see p4line15-28, Tab. 1

7. Coarse-grained plaque is misspelled.

R: This was corrected.

8. Some information comes later than expected, for example it is unclear which secondary (alexafuor594 or 647) is used for which primary (PHF-1 or RTN3) in the first description of immunofluorescence. This only becomes clear in the text about image analysis.

R: This was corrected in the methods.

9. Headings can be improved: the text about image analysis needs a different caption than what it currently is under (Immunohistochemistry).

R: This was corrected.

10. Not all figures that are referred to, can be found in the supplementary data: for example, I could not find Fig. S8, referred to in the section "Evaluation Metrics:".

R: This was corrected.

11. Figures are not always clear, for example
- figure 3(a): What does a VIP of 0.5 mean, is this the ratio of CP/DP?
- Figure 3(b): what do the cluster lines mean?

R: VIP is defined as Variable Importance in Projection indicating the effect size of the variable on the score that allowed separation of the plaque populations compared. The Figures were corrected and missing information further specified in the legends. Please note that the VIP plots are now in part in the SI.

Please see Fig 6 and SI Fig S2, S4 and S5

Other

12. The coarse-grained plaque is not the same as the primitive plaques or burned-out plaques. These are a different category, please correct this.

R: We appreciate this comment and removed this sentence.

13. The authors write about plaque maturation and fibrilization, but postmortem tissue is cross sectional, therefore the authors need to be aware this maturation is an assumption.

R: We concur and discuss this in part with cross-sectional/sequential data previously reported for eg Downs syndrome along with recent data using novel approaches such as stable isotope labelling. We further specify this in the limitations in the at the end of the discussion.

Please see p27line 25-p28line 5

Reviewer #3 (Remarks to the Author):

This is a well written paper that tries to shed light on the association of polymorphic A β plaque pathology with AD pathogenesis, clinical symptoms and disease progression. The authors are using advanced imaging tools such as functional amyloid microscopy and matrix assisted laser desorption ionization (MALDI) with TOF/TOF mass spectrometry. Because of the complexity of the data a considerable amount of statistical treatment of the data is done to try to draw out relationships between the observations from the data regarding the various forms of the A β plaques and their pathological relationships including using deep learning algorithms to interpret the mass spectrometry data.

We sincerely appreciate the very thorough and constructive review, particularly with respect to the MS experiments that helped to significantly improve the manuscript.

1. The authors need to define all abbreviations used in the paper including common abbreviations used in statistics. As an example, I couldn't find a definition for VIP score in the paper. Does it refer to variable importance plots?

Reply: We concur with this and specify all abbreviations in the manuscript. VIP refers to "Variable importance in Projection" which is updated in the text and the corresponding Figure legends (Figs 6 and SI Figs S2, S4, S5)

2. The authors state that they are using a Bruker rapifleX Tissue type MALDI TOF/TOF. I would have expected a better resolution than 4,500. Also, why were the samples run in linear mode when the TOF/TOF (reflectron) mode will give much better resolution and much improve mass spec data.

R: We appreciate this very important point and provide some clarification. We acquired data in liner pos. mode (LP) over reflective pos. mode (RP) mode due to superior sensitivity at high spatial resolution in linear mode. We demonstrate this in a separate Suppl. Figure (see Fig.S11 below). We further validated all peptide identities using ex situ IP-LC-MSMS. Those data are provided with in the SI (please see SI Fig S10).

3. Did the authors consider the possibility of **laser induced fragmentation** for some of the **A β** pieces observed?

R: We have been investigating the data and performed additional experiments with varying laser fluence in situ by spotting a Abeta 42 standard (100fmol) to evaluate any fragmentation under varying laser fluence settings. No in source fragmentation was observed in situ using the fluence settings applied for our MSI experiments (a) as outlined in the newly generated SI Fig S13.

4. The authors need to clearly note how many of the patient samples were analyzed using mass spectrometry. They report having 15 patient samples, 9 sporadic and 6 familial. How many tissue samples from each patient were analyzed using mass spectrometry?

R: We agree and have now specified this in the manuscript. We have expanded the cohort, and the number of patients was $n=23$ (12sAD, 6fAD, 5CUAP) and the number of replicates was $N=2$. This was updated in the methods and is specified in the respective Figure legends.

5. What thickness of tissue sample was used?

R: The cryosections were cut at 12 μm . This is now clarified in the methods.

6. In line 318-19 they state that serial sections were analyzed with both MALDI MSI and fluorescence microscopy.

Are the examples in part b and part c Figure 1 examples of specific tissue sections or aggregates of the serial sections.

R: We would like to clarify the experimental details as well as the images shown in Figure 1. For correlative MALDI and LCO imaging, data were acquired from the same tissue section where LCO imaging and deep learning classification guided MSI (Fig. 1B).

Consequently, the sections in Figure 1B are representative images for MSI and LCO data acquired from the same and not serial sections. This was corrected in the current revision.

In contrast, IHC experiments could not be performed on the same section. Here, the chromogenic IHC and HE images in Fig 1d (prev. 1b) are acquired from independent sections and serve merely the purpose of external pathological validation.

The fluorescent IHC and microscopy as shown in 1e (previously Fig 1C) was performed on serial sections from the same patients though solely for validation purposes.

We updated the figure, figure legends and results accordingly. See legend Figure 1; as well as Results: p7line 1-2

7., how many serial sections were there for each patient?

R: In general, we collected MALDI MSI data from N=2 serial sections per patient.

8. How reproducible were the mass specs from the serial sections?

R: We appreciate this important concern and evaluated the biological and technical variation for the MSI experiments.

Please see our response to Rev 1 regarding biological intra- and inter-patient variation (intra patient variation: <16%; inter patient variation: <18%.

In addition, we specifically evaluated technical variation by acquiring MALDI MSI data for an additional dataset of N=3 serial sections for one patient. Here the cumulative CV% was <10% across all technical replicates.

This information was included in the methods. P34line24-27

9. The authors need to clearly state what sample(s) were used for the MALDI MSI images presented in Figure 3 part c. If this was from a single tissue specimen, then the authors need to add a picture of the tissue so that readers can be oriented to the tissue morphology.

R: The MALDI MSI data shown in Figures 1, 3, 5 and 6 are representative data from one patient of the respective group (CUAP, sAD, fAD), including different plaque types captured from the same analyses of the same patient/tissue section to illustrate differences across plaque types.

We acknowledge that this needs to be clarified in a better way. For the data in Figure 3 we provide LCO and brightfield overlay images with the MSI data for better orientation across the tissue section (see SI Fig 2). We further specified the legend highlighting that these are representative data from one patient.

10. Also, the authors need to state the **mass window** used to generate the mass heat maps in part c and the mass of each of the polyforms should be included with the appropriate number of significant (digits) figures.

R: We appreciate this comment and have updated Figures S2, 3, 5 and 6 accordingly.

11. If the mass heat maps are all from the same tissue specimen, then the authors should include the full mass spectrum.

R: We acknowledge that more MS data need to be presented. We included plaque type specific, representative mass spectra in Figures 3, 5 and 6.

In addition, to the representative plaque ROI spectra for each plaque type we make all single-plaque data publicly available:

https://maciejdulewiczgu.shinyapps.io/Check_my_SpectRa/

12. This also applies to the single ion images presented in Figure 5. The authors need to clearly state the number of samples used.

R: We very much appreciate this important feedback. We specified in the legend that the ion images are representative from one patient each. We now further detail in the legend the number of replicates (n patients, N plaques) used for the statistical analysis.

13. The only mass spectrum shown in the paper is in figure 3 part h and it is averaged. The authors need to clearly state how the average mass spectrum being presented was generated. Were all the specimens used for the average mass spectrum?

R: We concur and provide details for the average MS spectra now included in Fig 3, 5, 6 and Fig S4 in the corresponding legend. In short the average spectra were generated from all plaques within one class for one patient. We however provide all the spectral data of all plaques across all groups:

https://maciejdulewiczgu.shinyapps.io/Check_my_SpectRa/

14. The images shown in general are not very clear in how they relate to tissue morphology which is the goal in mass spec imaging (MSI). One of the primary advantages of using MSI is to be able to relate the compounds being observed to the cellular structure of the tissue. The images show no discernible tissue structure.

R: We acknowledge this issue and would like to provide some explanation. The formic acid hydrolysis affects the signal of non-plaque deposited proteins (dispersion, degradation) and general tissue morphology making it difficult to visualize other regions based on ion signals.

Also, please note, as we only look in generally rather homogenous tissue (grey matter), outlining histological finer features, including different neural cell populations, with MALDI in protein mode is not the best suitable approach and lipid imaging would be more appropriate.

We however show correlative images with microscopy outlining general tissue structures for orientation (Fig. S1, Fig S2).

15. SI Table S1: Masses of the detected A β isoforms. The authors need to include at least representative mass spectra which show these isoforms.

R: We concur and provide average spectra of all plaque types across all groups in the Figs 3, 5 and 6 as well as in the SI: S1, S2 and S4. We further provide all raw spectra on a public server: https://maciejdulewiczgu.shinyapps.io/Check_my_SpectRa/

16. Were they observed in all MSI experiments.

R: The relevant amyloid isoforms that were compared across plaques were present in all the MSI experiments. We specified this in the methods (p p34line14-16) and results (p11line 13-14)

17. For each of the masses the exact mass for each one is calculated to 5 decimal places. Linear mode will not give the necessary resolution to justify this. What is being calculated for the [M+H]⁺ Avg? The values given for AVG look more like nominal masses. What is the purpose of presenting this data? The authors need to update the table using the proper number of decimal places for the masses and report the accurate mass data from the
the MSI experiments.

R: We appreciate this important comment and updated Table S1 accordingly showing the theoretical and detected Avg and monoisotopic values at the mass resolution/accuracy achievable in LP and RP mode. We also provide RP data in Fig. S11

18. The authors do not present a compelling argument for their conclusions using the mass spectrometry data. The mass spectrometry data does not give any insight into whether the A β polymorphism locations are coincidence or whether there is an actual correlation.

R: In order to address this important comment, we provide additional overlay data on LCO encoded plaque structure information and MALDI MSI single ion patterns. We show that indeed 1-40 localizes to the core of cored plaques in sAD, while 1-42 shows a more disperse pattern. This was demonstrated both in Figs 1, Fig 3 and a technical figure illustrating the spatial correlation of eg. Abeta 1-40 with the core. (see below) SI Fig S1b.

19. More careful mass spectrometry work needs to be done rather than applying a battery of statistical tools.

R: We concur and provide additional data addressing the various points regarding to MSI analysis specifically: mass- vs. spatial resolution (Fig S11), laser fluence effects, and signal enhancement with FA(Fig S12). In addition we performed ex situ IP-LC-MSMS analyses for peptide sequence validation (Fig. S10).

20. I also believe that this holds true for the microscopy data for similar reasons as stated above for the mass spectrometry data. It is unclear what samples are being used.

R: We appreciate this comment and provide the number of samples analyzed for all results in the corresponding Figure legends.

Please see Fig 4.

21. MSI is used ideally with no specific physiological outcome in mind but being untargeted allowing the biochemistry to shine through. I would expect many other small proteins and peptides to be observed in this mass region (1500 to 6000 Da).

R: Typically, this holds true for peptide imaging. However, we use FA hydrolysis to enhance the Abeta signal. This has however, at least in our hands, the limitation that other endogenous signals are decreased and dispersed. To demonstrate this effect of hydrolysis on the Abeta signal, we prepared additional experiments with and without FA treatment.

Please see supplemental Figure S12.

(B) After hydrolysis

Without hydrolysis

Reviewer #4 (Remarks to the Author):

This work displays effective use of machine learning and deep learning to discover new chemical insights into AD, with extendability to other spectroscopy/MSI tasks and samples. The discussion clearly describes the tangible findings, and statistics are reported to support claims. However, revisions should be made surrounding clarity/transparency of the methods, and importantly the disclosure of code and workflows (i.e. a GitHub), as currently this work is non-reproducible. Additionally, ML terminology can be standardized when appropriate.

1. Line 121: Briefly explaining why MSI provides an advantage over other techniques would be beneficial, or potentially mentioning how MALDI-MS can quantize various analytes simultaneously with high throughput, etc. rather than stating its one of many modalities, superior to staining, would better provide readers with the context of why the particular method was employed. It may also be beneficial to describe MSI as being an orthogonal approach to staining/spectroscopy since immuno-staining was done.

R: We appreciate this comment and address this in the introduction. We further generate a technical SI Figure (Fig S1) highlighting the MSI capabilities.

2. Could also make sense to have some text and citations regarding DL for image classification and/or MSI in the context of histology etc.

R: Thank you for this feedback. We have included the following paragraph in the introduction of the manuscript.

Deep learning has emerged as a transformative approach in the field of MSI, enabling advanced analysis and interpretation of complex biological samples. Recent studies have highlighted the efficacy of deep learning techniques in enhancing image classification accuracy and facilitating the identification of biomarkers in histological contexts (Amitay et al., 2023; van der Laak et al., 2021; von Chamier et al., 2021). In this study, we apply a deep learning model to classify MSI plaque images, contributing to the growing body of research that seeks to harness these powerful techniques for enhanced histopathological analysis.

Please see p5line24-28

3. Line 190: The actual architecture should be reported as code for training and with model weights for final inference. Important details such as image depth (i.e. multiple wavelength channels) that are not made explicit in text would be covered by accompanying code (as the figures of the architecture are not comprehensive, nor should they). Details such as the loss function, mini-batch size, learning rate (and decay) should be reported, as well as loss vs epoch plots in the supporting.

R: Thank you for your insightful comments. We have addressed your concerns as follows in the methods:

Model Architecture and Code:

The code and trained model weights are uploaded on to Github and will be made available upon reasonable request. Interested readers or researchers can contact the corresponding author for access. We have included a statement in the relevant section. However, we provide the code for review purposes as mandated by the journal publisher: Please see: https://github.com/meerask16/plaque_classifier

Further clarity on model training:

We have rewritten the relevant text on the manuscript to provide further clarity to the mentioned aspects. Please find attached the relevant section below that was included in the Methods:

“Data Collection and Preprocessing: A dataset of LCO-based fluorescent amyloid plaque images was collected, ensuring a diverse representation of plaque types. Annotations of the amyloid plaque images were classified into coarse, cored, and diffused classes based on morphology. The model was trained and tested using a dataset of annotated amyloid plaque images (n = 385), consisting of Coarse grain plaques (n = 42), Cored plaques (n = 208), and Diffused plaques (n = 135). The images were preprocessed to standardize size (120×120), adjust contrast, and convert the images to a depth of 3 wavelength channels, corresponding to 120×120×1. This preprocessing helped minimize variations that could affect model performance. The input image resolution and multiple wavelength channel details are also documented in the code, and available upon request.

Network Architecture: We designed a fully connected neural network consisting of multiple convolutional layers followed by max-pooling layers for feature extraction. Batch normalisation and dropout layers were employed to enhance model generalisation and prevent overfitting. The final classification layer utilized softmax activation to output probabilities for each plaque type.

Training Procedure:

Three-fold cross-validation: From the 385 images, 20% (n = 78) were set aside as an unseen test dataset. The remaining 307 images were used for training and validation. In each iteration of the cross-validation process, the dataset is split into three mutually

exclusive subsets. Two subsets (80% of the data) are utilized for training, while the remaining subset (20%) serves as the validation set. Importantly, no data points from the validation set are included in the corresponding training set for that iteration, ensuring that each image is used for validation only once. The predictions of the three trained models are averaged to create a final model. This approach enhances the robustness of the final model by preventing any overlap between training and validation sets.

Model training:

The classification model was trained on each combination of training and validation subsets, repeated three times, covering all possible combinations. We employed the Adam optimizer with a learning rate of 0.001 and a learning rate decay schedule (decay factor of 0.96 every 10 epochs) to ensure efficient convergence towards optimal weights. The loss function used was categorical cross-entropy, and we used a mini-batch size of 32 for training. Data augmentation techniques, such as random rotation, flipping, and scaling, were applied to augment the training set, improving model robustness and reducing overfitting.

Code Availability Statement:

Due to confidentiality and transparency considerations, the full code and model weights cannot be shared publicly. However, they can be made available upon reasonable request to the corresponding author for research purposes in the interest of fostering scientific collaboration, transparency and reproducibility of our research.“

4. Line 196: Its not clear why cross-training is being used as terminology instead of cross-validation.

R: Thank you for pointing this out. We agree that the term "cross-training" may be confusing. The correct term should indeed be "cross-validation." We have updated the manuscript accordingly to reflect the standard terminology of cross-validation.

5. Line 199 Is it the parameters or predictions that are averaged? There should be an accompanying code for the deep learning. Would the authors be employing an ensemble model as their 'final model'?

R: Thank you for these valuable remarks. To clarify, predictions are averaged across the models, not the parameters.

We have revised and addressed this point in the manuscript accordingly. Please see p32line5

6. Line 203: Do the authors mean each pair of training and validation splits? Each combination would be highly irregular, as there would certainly be identical data in both the training and validation for many combinations.

R: Thank you for raising this important issue. To clarify, we have performed a standard cross-validation procedure in which each fold is split into mutually exclusive training and validation sets. In our three-fold cross-validation approach, no data points from the validation set are included in the corresponding training set. This ensures that each image

is used for validation only once per iteration, preventing overlap between the training and validation sets in each fold.

7. Line 213: The SI only goes to page 10 and ends with Figure S7. Table of contents also states SI Figures S1-7

R. We concur and significantly expanded and updated the Figures and SI Figures. The correct order and cross references are now updated in the main text.

8. Line 234: Authors should make clear to readers how HCA was employed, potentially in the SI. Please change 'k mean' to 'k-means'. Code regarding HCA should be available. Is pseudo coloring being done on a pixel by pixel basis by pseudo cluster assignment?

R: We appreciate this thorough comment. We performed several multivariate analyses incl. two kinds of cluster analysis as specified below.

a) Cluster analyses for top down MSI data segmentation

b) OPLSA-DA for single spectral data analysis followed

c) HCA based clustering of averaged peptide intensities across plaque types and groups
d) peptide regression analyses within distinct plaque types in yielding a similar information as the clustered correlation and intensity maps previously included.

For image segmentation of MSI data (a) we used a commercial software (SciLLS Pro v 2024, SciLLSLab, Bremen, Germany) with limited information on the exact model beyond k-means. (We have corrected the terminology from "k mean" to "k-means" in the manuscript).

Regarding the cluster analysis code no details were made available by the vendor. The pseudo-colouring was done by the software on a pixel to pixel basis. We have specified the few available parameters (normalization, denoising) used for image segmentation and refer to key references of the developers. This was specified in the methods.

Hierarchical cluster analysis of ROI spectral data, was performed with Metaboanalyst, again with limited information on the code but with the following parameters were: Data: normalized, Standardization: Autoscale features; Clustering method: Ward, Distance: Euclidian; Colour contrast: default.

This has all been updated in the methods. Here, we specified the available parameters that were selected in the methods along with key references. p33line16-p34line19

9. Supporting figures only show AB isoform intensities as variables for the HCA, and it would not be clear to many readers how this can lead to segmentation. Any manual assessment of initial clustering/ML required for setting the final parameters for tasks such as dendrogram cuts should be described if employed.

R: We assume that this refers to the segmentation shown in Fig1b and agree that this needs more detail. The segmentation workflow was performed in SciLLS using bisecting k-means CA where different subclusters are represented by different colors providing some visual representation of the chemical differences encoded in different pseudoclusters. Any manual assessment was based on the LCO overlay stratifying the

pseudoclusters towards different plaque types identified in the ML-LCO microscopy approach. Dendrogram cuts were carried out in SciLLS to identify whether the segmentation can distinguish those plaques. Of note this is just one initial approach to investigate the MSI data comprise the necessary chemical diversity to differentiate plaques warranting further bottom-up analysis as SciLLS is limited in comparing multiple data sets across different groups. Consequently, the large body of statistical analysis in the manuscript is based on ML-LCO annotation, plaque ROI selection (outlined by the ML) and extraction of ROI MS spectral data.

We updated Fig 1 and the corresponding legend and detail this in the methods (p34line3-7) and results (p9line 9-15)

10. Line 339: Would be interesting to have label probabilities for these misclassified samples to gain insight into the model. Did these misclassifications make visual sense given similarity to other classes? In any case, AUC and ROC curves should be reported.

R: Thank you for your valuable feedback. We agree that including label probabilities for misclassified samples would provide valuable insights into the model's behavior. To address this, we have added a supplementary figure and examples of misclassified samples, along with their associated label probabilities. While we acknowledge the importance of AUC and ROC curves, they are not directly applicable to multi-class classification tasks. Instead, we have employed metrics such as accuracy, precision, recall, and F1 score for each class to evaluate model performance.

We agree that providing label probabilities for misclassified samples would enhance understanding of the model's performance. Hence, we have included a supplementary table with a few examples of misclassified samples. See SI Tab.2

Misclassified Images	True Class	Predicted Class
	Coarse $p = 0.38$	Cored $p = 0.42$
	Cored $p = 0.47$	Coarse $p = 0.51$
	Diffused $p = 0.48$	Cored $p = 0.51$

AUC and ROC Curves:

Regarding the request for AUC and ROC curves, we acknowledge their importance for evaluating model performance. However, for multi-class classification, traditional AUC/ROC curves are not directly applicable. Instead, we focused on other evaluation metrics, such as accuracy, precision, recall, and F1 score for each class. We will clarify this limitation in the manuscript and discuss alternative evaluation strategies that can be employed for multi-class settings.

11. Line 348: These are indeed great results. Was the model benchmarked against other (plaque) segmentation models? This is a very low amount of data for deep learning, was transfer learning not employed?

R: Thank you for your insightful observations.

Benchmarking Against Other Models:

We acknowledge the importance of benchmarking against existing plaque segmentation models. However, this study primarily focused on the initial development and validation of our model.

We have included a statement in the manuscript to clarify that future work will involve benchmarking against state-of-the-art segmentation models to provide greater context and validation for our approach.

Transfer Learning:

It is correct that the amount of data available for deep learning is limited. In this proof-of-concept study, we did not employ transfer learning due to the specific nature of our dataset and the preliminary nature of our model development. However, we recognize the potential benefits of transfer learning, especially with a small dataset, and plan to explore this approach in subsequent phases of our research as we aim to expand our dataset and model complexity.

Please see p29 line 13-15

12. Figure 2e seems unnecessary given its description in text.

R: The figure was revised accordingly.

Reviewer #4 (Remarks to the Author):

The authors have done a wonderful job addressing many of the requested revisions and questions.

They have shown excellent work and given the availability of their trained model I leave acceptance of the article to the editors.

Reviewer #4 (Remarks on code availability):

The code is easy to follow, and well organized. Training scripts should be provided, however.

We appreciate the positive feedback. Training scripts are provided at the Github repository together with the code.

https://github.com/meerask16/plaque_classifier

We specify the location of the code in the Code Availability Statement in the manuscript. (please see p29)